# SCALING MULTI-AGENT ENVIRONMENT CO-DESIGN WITH DIFFUSION MODELS

## ABSTRACT

The *agent-environment co-design* paradigm jointly optimises agent policies and environment configurations in search of improved system performance. With application domains ranging from warehouse logistics to windfarm management, co-design promises to fundamentally change how we deploy multi-agent systems. However, current co-design methods struggle to scale. They collapse under high-dimensional environment design spaces and suffer from sample inefficiency when addressing moving targets inherent to joint optimisation. We address these challenges by developing **Diffusion Co-Design (DiCoDe)**, a scalable and sample-efficient co-design framework pushing co-design towards practically relevant settings. DiCoDe incorporates two core innovations. First, we introduce Projected Universal Guidance (PUG), a sampling technique that enables DiCoDe to explore a distribution of reward-maximising environments while satisfying hard constraints such as spatial separation between obstacles. Second, we devise a critic distillation mechanism to share knowledge from the reinforcement learning critic, ensuring that the guided diffusion model adapts to evolving agent policies using a dense and up-to-date learning signal. Together, these improvements lead to superior environment-policy pairs when validated on challenging multi-agent environment co-design benchmarks including warehouse automation, multi-agent pathfinding and wind farm optimisation. Our method consistently exceeds the state-of-the-art, achieving, for example, 39% higher rewards in the warehouse setting with 66% fewer simulation samples. This sets a new standard in agent-environment co-design, and is a stepping stone towards reaping the rewards of co-design in real world domains.

## 1 INTRODUCTION

The performance of agents is fundamentally tied to the environments they inhabit. In real world settings, engineers have many opportunities to coordinate agent policies and environments together. For example, contractors match robot delivery policies with an ordered grids of shelves to streamline deliveries in autonomous warehouses (Christianos et al., 2020) and energy engineers strategically control the placement of turbines to maximise energy capture in wind farms (Bizon Monroc et al., 2024). Agent-environment co-design is a paradigm that captures this coupling by jointly optimising environments $\theta$ and agent policies $\pi$ for a shared goal, with the potential to fundamentally reshape how we deploy multi-agent systems by enabling performance gains unachievable with tuning either agents or environments alone. Agent-environment co-design has attracted much interest in recent years, with theoretical results establishing a link between the existence of efficient policies and the choice of environment (Amir & Bruckstein, 2025), and existing methods producing successful agent-environment pairs using RL (Cheney et al., 2018; Schaff et al., 2019; Gao & Prorok, 2023). However, these methods hit a scalability barrier when faced with high-dimensional environments thus restricting their application to toy problems. We identify two fundamental obstacles:

**1. The Curse of Combinatorial Design Spaces.** Real-world environments often comprise numerous elements with domain-specific constraints, inducing an exponential explosion in possibilities. For example, placing 50 obstacles on a $16 \times 16$ grid yields $\binom{256}{50} \approx 10^{53}$ configurations. Conventional approaches struggle. Methods relying on simple distributions, such as truncated Gaussians (Gao & Prorok, 2023), impose restrictive assumptions and lack the expressivity to capture com-

plex environmental structures. Evolutionary methods (Cheney et al., 2018) scale poorly with design dimensions, and sequential generators (Dennis et al., 2020) impose an unsuitable temporal structure.

**2. Sample Inefficiency Driven by Policy Shift.** As agent policies evolve during training, the optimal environment shifts (Van Hasselt et al., 2018), a phenomenon we refer to as policy shift. Existing methods typically address this by freezing the policy while updating the environment generator. This approach is highly sample-inefficient, as the decoupled optimisation prevents shared utilisation of costly rollout data. Moreover, the scalar episode return is often used as the sole learning signal for the environment generator, discarding valuable information contained within the trajectory.

To overcome these limitations, we propose **Diffusion Co-Design (DiCoDe)**, a scalable and sample-efficient framework that harnesses the power of guided diffusion models and multi-agent reinforcement learning (MARL). Diffusion models have emerged as the state-of-the-art for modelling complex, high-dimensional distributions (Dhariwal & Nichol, 2021). Although recently validated in the distinct area of unsupervised environment design (UED) (Chung et al., 2024), their potential for the cooperative co-design problem remains largely untapped. DiCoDe introduces two key innovations:

*Projected Universal Guidance (PUG) for Constrained Environment Generation.* To navigate complex design spaces, we develop PUG, a novel sampling technique unifying universal guidance (Bansal et al., 2023) with projective constraints (Christopher et al., 2024). PUG generates high-rewarding environments while enforcing hard physical constraints (e.g., non spatial overlap), significantly improving the quality of generated designs compared to standard classifier guidance.

*Critic Distillation for Knowledge Sharing.* To address sample inefficiency, we break the separation between agent training and environment optimization. Instead of treating agent training as a black box, DiCoDe employs a distillation mechanism to explicitly share knowledge from the MARL critic directly into an environment critic used to guide the diffusion model. This provides a dense, low-variance, and up-to-date learning signal for the environment generator without freezing agent or environment generation policy at any point, thus drastically reducing the need for costly simulation rollouts and rapidly adapting the environment generator to the current agent capabilities.

We evaluate DiCoDe on a suite of challenging multi-agent co-design scenarios adapted from established benchmarks in warehouse management (D-RWARE) (Christianos et al., 2020), wind farm control (WFCRL) (Bizon Monroc et al., 2024), and multi-agent pathfinding (VMAS) (Bettini et al., 2022). Our experimental results demonstrate that DiCoDe significantly outperforms existing co-design methodologies, discovering environment-policy pairs that improve task rewards by up to 39% while achieving a 66% reduction in sample complexity.

## 2 PRELIMINARIES

We briefly describe underspecified games and diffusion models, the foundations of our work.

### 2.1 ENVIRONMENT CO-DESIGN OVER UNDERSPECIFIED GAMES

The co-design problem can be formalised as an underspecified (Dennis et al., 2020) RL problem. We adopt the formulation by Samvelyan et al. (2023) to account for designable multi-agent environments. Consider an underspecified partially observable stochastic game (UPOSG). $\langle n, \mathcal{A}, \mathcal{O}, \mathcal{S}, \mathcal{P}, \mathbf{\Omega}, \mathcal{R}, \gamma \rangle$ with $n$ agents. Let subscripts denote the timestep: trajectories $\tau = s_0, \boldsymbol{a}_0, \boldsymbol{r}_0, s_1, \boldsymbol{a}_1, \boldsymbol{r}_1 \dots$ are drawn with states $s_t \in \mathcal{S}$ and actions $\boldsymbol{a}_t \in \mathcal{A}$ (joint action space $\mathcal{A} = \{\mathcal{A}_1, \mathcal{A}_2, \dots, \mathcal{A}_n\}$). $\mathcal{O} = \{\mathcal{O}_1, \mathcal{O}_2, \dots, \mathcal{O}_n\}$ denotes the joint observation space of the $n$ agents; $\mathbf{\Omega} = \{\Omega_1, \Omega_2, \dots, \Omega_n\}$ are the respective observation functions of each agent where $\Omega_i$ is a function $\mathcal{S} \to \mathcal{O}_i$. Finally, design space $\Theta$ may refer to the space of object layouts or physical dynamics, inducing a conditioned transition function $\mathcal{P}_\theta(s_{t+1}|s_t, a_t)$ and initial state distribution $\mathcal{P}_\theta(s_0)$. We defined environment instantiation $\mathcal{E}$ as the function from $\theta$ to $s_0$. The agent objective is captured by the reward function $\mathcal{R} : \mathcal{S} \times \mathcal{A} \to \mathbb{R}^n$ supplying rewards $r_t^i$, superscript to denote agent index.. We assume agents are collaborative, and the team objective is to maximise the sum of agent rewards. The co-design objective is an optimal tuple $(\theta^\star, \phi^\star)$ such that the agents are able to effectively complete their tasks in environment $\theta^\star$ under $\phi^\star$ parameterised policy $\boldsymbol{\pi}_{\phi^\star} : \mathcal{O} \to \mathcal{A}$. Formally, this goal is captured as

$$J(\phi, \theta) = \mathbb{E}_{\tau \sim (\boldsymbol{\pi}_\phi, \theta)} \left[ \sum_{i=1}^{n} \sum_{t=0}^{\infty} \gamma^t r_t^i \right] \qquad (\phi^\star, \theta^\star) = \underset{\theta \in \Theta, \, \phi \in \Phi}{\arg\max} \, J(\phi, \theta) \quad . \qquad (1)$$

## 2.2 GUIDED DIFFUSION MODELS

At a high level, a diffusion process (Ho et al., 2020; Dhariwal & Nichol, 2021; Bansal et al., 2023; Christopher et al., 2024) iteratively adds noise to a sample $x_0$. This may be represented as a variance preserving (VP) stochastic differential equation (SDE) (Song & Ermon, 2019; Song et al., 2021b) with standard Brownian motion $w$ and noise schedule $\beta$ evolving over time $t$

$$dx = -\tfrac{1}{2}\beta(t)x dt + \sqrt{\beta(t)} dw. \qquad (2)$$

A famous result by Anderson (1982) (applied to the VP SDE) states that the reverse process is given by the reverse-time SDE:

$$dx = -\beta(t) \left[ \tfrac{1}{2}x + \nabla_x \log p(x; t) \right] dt + \sqrt{\beta(t)} d\overline{w}. \qquad (3)$$

Here, $\overline{w}$ stands for the reverse process of $w$. Therefore, given a score function $\log p(x_t, t)$, such as a neural network $\varepsilon_\varphi$ parameterised by $\varphi$ and trained with score-matching (Song & Ermon, 2019), it is possible to sample $x_0$ by following the reverse SDE with initial condition of $x_T = \mathcal{N}(0, \mathbf{I})$. For example, DDIM (Song et al., 2021a) may be considered a compute efficient discretisation of the VP SDE. Given a noise schedule $\alpha_0 = 1, \ \alpha_t = \alpha_{t-1}(1 - \beta_t), \ t = 1, \ldots, T$, this is represented as

$$x_t = \sqrt{\alpha_t}x_0 + \sqrt{1 - \alpha_t}\epsilon, \epsilon \sim \mathcal{N}(0, \mathbf{I}). \qquad (4)$$

We may also be inclined to sample from a conditional distribution $p(x_0|y)$, where $y$ is a condition (e.g. environment description). In the co-design process, this allows us to specify specific properties of desired environments. The score function in Equation 3 may be decomposed conditionally as

$$\nabla_{x_t} \log p(x_t|y; t) \propto \nabla_{x_t} \log p(x_t, y; t) = \nabla_{x_t} \log p(y|x_t; t) + \nabla_{x_t} \log p(x_t; t). \qquad (5)$$

Dhariwal & Nichol (2021) introduce *classifier guidance* by learning a time-dependent classifier $c_\vartheta(y|x_t, t)$. The gradient of $\vartheta$ wrt. $x_t$ is an approximation of $\log p(y|x_t; t)$ and $\nabla_x \log p(x_t; t)$ is the score function of the unconditional diffusion process. Christopher et al. (2024) further introduce projected diffusion models (PDM) as a method to enforce constraints in the process, and Bansal et al. (2023) propose universal guidance to improve the quality of generated conditional samples beyond classifier guidance. Additional details of diffusion models are provided in Appendix A.1.

## 3 RELATED WORK

In prior co-design literature, Cheney et al. (2018) apply evolutionary methods to the morphology of robots, whereas Hauser (2013) remove navigation obstacles. Roodbergen et al. (2015) jointly design warehouse control policies and layouts. Zhang et al. (2024) scale optimisation of cellular warehouses using agent-based simulations, but do not train the robot control policy. Jain et al. (2017) incorporate the environment (dynamic cache partitioning) as part of the POSG and leverage MARL to train agents. Schaff et al. (2019) transform environment (robot morphology) design into a reinforcement learning problem and apply policy gradient. Gao & Prorok (2023) formalize the co-design process and coordinate the optimization of environment generation and agent policies in a mutually recursive process with MAPPO and policy gradient. Compared to simpler representations such as truncated Gaussians (Gao & Prorok, 2023), Gaussian mixture models (Schaff et al., 2019) or binary decisions (Hauser, 2013), our work is the first to leverage diffusion models for co-design, enabling scaling to high-dimensional domains. It is also the first learning co-design method to distil knowledge between agents and environment, explicitly addressing sample inefficiency and moving targets. Moreover, we evaluate over general domains without restriction to a certain class of co-design scenarios.

Unsupervised environment design (UED) is a related area of research with a distinct focus on curriculum training. Under dual curriculum design (DCD) (Jiang et al., 2021a), an agent policy is trained with RL against an adversarial environment generator. Dennis et al. (2020) employ a RL approach with environment learnability as reward, whereas Jiang et al. (2021b;a) prioritise level

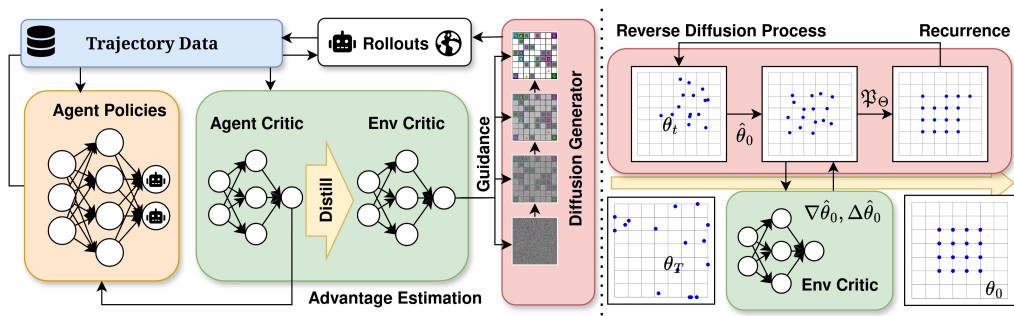

Figure 1: General framework of our diffusion co-design method. In extension of a MARL iteration, we introduce an environment critic trained using critic distillation. This guides a diffusion model via a carefully designed sampling process that satisfies hard constraints, generating a distribution of highly-rewarding environments to collect trajectories upon. Repeating this process leads to consistently superior policy-environment tuples.

replay (PLR) from a uniform generator. Parker-Holder et al. (2022) successfully combine evolutionary methods and PLR with ACCEL. (Samvelyan et al., 2023) extend UED to the multi-agent game in MAESTRO. Chung et al. (2024) introduce the use of diffusion models in the UED domain (ADD). By proposing a differentiable measurement of regret, they are able to exploit classifier guidance on pre-trained diffusion models to both produce meaningful environments and maintain the diversity of generated environments. Although UED is a fundamentally different paradigm to co-design with conflicting rather than shared objectives, lessons on information architectures can be shared. Building upon the codebase of regret-guided diffusion models in ADD, we develop a novel sampling technique for constraint-aware environment diffusion broadly applicable to UED as well as co-design.

## 4 METHODOLOGY

We develop **Diffusion Co-Design** (DiCoDe) (Figure 1) as a sample-efficient and scalable framework for multi-agent environment co-design by harnessing critic-guided diffusion models. At a high-level (Figure 1), DiCode consists of several delineated components. First, DiCoDe pre-trains a diffusion model $\epsilon_\varphi$ on a uniform exploration distribution. Then, in the main training loop, DiCoDe alternates between sampling environments, executing rollouts, and updating parameters for agent policy $\pi_\phi$ or environment critic $\mathcal{V}_\vartheta$. Crucially, environments are drawn from a reward-maximising distribution (Section 4.1) using a novel guidance method tailored for environment generation (Section 4.2). We adopt multi-agent proximal policy optimisation (PPO) (Yu et al., 2022) as the underlying RL engine to optimise $\phi$, and introduce a knowledge sharing distillation mechanism (Section 4.3) to efficiently update $\vartheta$. We conclude with comments on the overall framework and its advantages in Section 4.4.

### 4.1 EXPLORING PERFORMANT ENVIRONMENTS WITH GUIDED DIFFUSION

A pillar of co-design is a desirable distribution over environments. Ideally, this distribution should exploit the current policy behaviour to achieve a high reward and explore the space of environments to avoid local optima. We define the soft co-design distribution $\Lambda_\phi^\star$ to maximise

$$\Lambda_\phi^\star = \arg\max_\Lambda \left[ \mathbb{E}_{\theta\sim\Lambda}\left[J(\phi,\theta)\right] + \tfrac{1}{\omega}H(\Lambda) \right], \tag{6}$$

where $\omega$ is a weighting hyper-parameter and $H(\Lambda) = -\sum_{\theta\in\Theta}\Lambda(\theta)\log\Lambda(\theta)$ is the entropy of distribution $\Lambda$. We can interpret the entropy bonus as a regularisation term to encourage exploration of the environment space, akin to the entropy regularisation term in RL (Schulman et al., 2017). The solution to $\Lambda_\phi^\star$ is a well-known result (Jaynes, 1957), with score

$$\nabla_{\theta_t}\log\Lambda_{\phi,t}^\star(\theta_t) \propto \nabla_{\theta_t}u_t(\theta_t) + \omega\nabla_{\theta_t}J_t(\phi,\theta_t) \tag{7}$$

where $t$ is the diffusion time-step and $\theta_t$ is the environment diffused by the forward process. $u$ is the uniform exploration distribution, and we subscript $u, J$ with $t$ to denote time-dependent values: $u_t(\theta_t) = u(\theta_0)$ and $J_t(\phi, \theta_t) = J(\phi, \theta_0)$.

It is possible to approximate $\nabla_{\theta_t} u_t(\theta_t)$ with a pre-trained diffusion model $\varepsilon_\varphi$, or equivalently $\epsilon_\varphi = -\sqrt{1 - \alpha_t}\varepsilon_\varphi$, assuming access to a procedural environment generator to sample from $u$. Therefore, given an environment critic $\mathcal{V}'_\vartheta : \Theta \times \mathbb{N} \to \mathbb{R}$ trained to approximate environment returns $J_t(\phi, \theta_t)$, we can formulate a reverse diffusion sampling process by substituting Equation 7 into Equation 3.

$$d\theta_t = -\beta(t) \left[\tfrac{1}{2}\theta_t + (\nabla_{\theta_t} u_t(\theta_t) + \omega \nabla_{\theta_t} \mathcal{V}'_\vartheta(\theta_t, t))\right] dt + \sqrt{\beta(t)}d\overline{w} \tag{8}$$

In prior UED literature for environment generation using diffusion (Chung et al., 2024), the reverse process is sampled with DDIM and $\mathcal{V}'_\vartheta$ trained on noise-injected environments to condition a *time-dependent* critic. However, we find empirically that $\mathcal{V}'_\vartheta$ is not effective at estimating the reward of noise-injected environments. We speculate this is due to low signal-to-noise ratio induced from noisy $\theta_t$ combined with aleatoric uncertainty of environment returns. Additionally, the pre-trained diffusion model inadequately constrains the diffusion process, leading to invalid environments when $\omega$ is increased because $\theta_t$ leaves the data manifold.

## 4.2 Projected Universal Guidance

To overcome the limitations (Section 4.1) of standard classifier-guidance in environment generation, we propose projected universal guidance (PUG) as an unification of universal guidance with PDM. First, we incorporate the insight that the expected clean image

$$\hat{x}_0^t = \epsilon'_\varphi(x_t, t) = \tfrac{1}{\sqrt{\alpha_t}}\left(x_t - \sqrt{1 - \alpha_t}\epsilon_\varphi(x_t, t)\right) \tag{9}$$

is a suitable input for an environment critic via direct application of universal guidance (Appendix A.2). Consequently, we can replace $\mathcal{V}'_\vartheta$ with an environment critic $\mathcal{V}_\vartheta$ trained directly on environments $\theta_0 = \theta$ predicting the expected return.

Second, consider the scenario design space $\Theta$ as a feasible region within a wider diffusion domain $\Theta \subseteq \boldsymbol{X}$ and that $\epsilon_\varphi, V_\vartheta$ operate on the wider do-

**Algorithm 1:** Projected Universal Guidance (PUG)

---
**Input:** $k, m, \omega, \mathfrak{P}_\Theta, V_\vartheta$
**for** $t = T, T-1, \ldots, 1$ **do**
    **for** $n = 1, 2, \ldots, k$ **do**
        $\hat{\theta}_0 \leftarrow$ Equation 9 composed with $\mathfrak{P}_\Theta$;
        $\hat{\epsilon}_{\varphi,\vartheta}(\theta_t, t) \leftarrow$ Equation 17;
        **for** $n = 1$ **to** $m$ **do**
            $\overline{\epsilon}_{\varphi,\vartheta}(\theta_t, t) \leftarrow$ as Equation 19
        Compute $\tilde{\epsilon}_{\varphi,\vartheta}(\theta_t, t) \leftarrow \mathfrak{P}_\Theta(\overline{\epsilon}_{\varphi,\vartheta}(\theta_t, t), \theta_t, t)$;
        $\theta_t \leftarrow$ Equation 20 with $\tilde{\epsilon}_{\varphi,\vartheta}(\theta_t, t)$;
    Sample $\theta_{t-1}$ using the diffusion process;
**return** generated sample $\theta_0$;

---

main $\boldsymbol{X}$. For example, $\Theta$ may be the set of images identifying an environment and $\boldsymbol{X} = \mathbb{R}^{H \times W \times 3}$. Our goal is to constrain all generated samples to be in $\Theta$, assuming there exists a projection operator $\mathfrak{P}_\Theta : \boldsymbol{X} \to \Theta$ that maps a sample $x \in \boldsymbol{X}$ to the closest valid environment $\mathfrak{P}_\Theta(x)$. We overload the definition of $\mathfrak{P}_\Theta$ to be applied to noise.

$$\mathfrak{P}_\Theta(\epsilon, \theta_t, t) = \tfrac{1}{\sqrt{1-\alpha_t}}x_t - \tfrac{\sqrt{\alpha_t}}{\sqrt{1-\alpha_t}}\mathfrak{P}_\Theta(\epsilon'(\theta_t, t), \theta_t, t) \tag{10}$$

Our proposed PUG applies $\mathfrak{P}_\Theta$ onto the predicted clean image in the universal guidance process to enforce constraints. The complete algorithm is shown in Algorithm 1.

Compared to PDM, our method does not require $\theta_t \in \Theta$ thereby relaxing unnecessary constraints within the diffusion process. PUG generates high-quality environments with DDIM as the underlying diffusion process, whereas Christopher et al. (2024) found that PDM exhibited suboptimal performance with DDIM.

## 4.3 Learning an Environment Critic

Recall the learning target of the environment critic $V_\vartheta(\theta) \to^{\text{train}} J(\phi, \theta)$. We remark the UPOSG can be viewed as an equivalent POSG where the first state-action pair is environment generation, all later states includes $\theta$, and the environment generator is a separate agent acting on the first state-action pair (Simaan & Cruz Jr, 1973). In this formulation, $J(\phi, \theta)$ is closely related to the value function $V^{\boldsymbol{\pi}}(s_t) = \mathbb{E}_{\tau \sim \boldsymbol{\pi}}\left[\sum_{i=0}^\infty \gamma^i r_{t+i}\right]$ used in RL algorithms to obtain the expected return. Agent critics

are estimators of the value function, typically used to reduce variance (Sutton et al., 1999) or obtain the policy directly (Mnih et al., 2015). In our use case, a standard agent critic is a promising surrogate target for the environment critic. Suppose the agent critic is an unbiased estimator, then:

$$J(\phi, \theta) = \mathbb{E}_{s_0 \sim \mathcal{P}_\theta} [V(s_0)] = \mathbb{E} [\mathbb{E}_{s_0 \sim \mathcal{P}_\theta} [V_\psi(s_0)]]. \tag{11}$$

There are three clear advantages to using an environment critic extracted from the agent critic. First, the agent critic is trained on all transition tuples $(s_t, a_t, r_t, s_{t+1})$ collected, which is more informative than just the sampled episode return $J(\phi, \theta)$ used by previous methods. Additionally, because the agent critic is trained jointly on the same data as the agent policy (with off-policy adaptations (Mnih et al., 2016) predetermined by the RL algorithm), we can assume the agent critic adapts to the current policy. Distilling this to the environment critic mitigates policy-shift with an accurate and up-to-date signal. Third, the agent critic provides targets with low variance by filtering out stochasticity within an episode from the policy or transition function, which we hope may improve training stability.

It is possible to leverage knowledge of the environment design space to assist in constructing the environment critic. If $\mathcal{E}$ is differentiable, we may backpropagate through $\mathcal{E}$ to directly use the agent critic as an environment critic. If not, we propose to train the environment critic on a distillation loss

$$\mathcal{L}_{\text{distill}}(\vartheta, \boldsymbol{\theta}) = \sum_{\theta \in \boldsymbol{\theta}} (\mathcal{V}_\vartheta(\theta) - \mathbb{E}_{s_0 \sim \mathcal{P}_\theta} [V_\psi(s_0)])^2 \tag{12}$$

using Monte-Carlo sampling to estimate $\mathbb{E}_{s_0 \sim \mathcal{P}_\theta} [V_\psi(s_0)]$ with $M_{\text{distill}}$ samples. Choosing a suitable $M_{\text{distill}}$ balances between variance reduction (due to $s_0$) and computation speed. $\boldsymbol{\theta}$ is a design choice for the practitioner: we suggest sampling from a FIFO memory buffer $\mathcal{D}$ of the previous $N$ environments used to train the agent, but it is also possible to sample $\theta$ on demand by calling PUG. We describe this process as *distillation* due to similarities with knowledge distillation literature (Hinton et al., 2015). Certainly, any techniques there will apply to our setting.

### 4.4 DIFFUSION CO-DESIGN (DICODE)

We now present the full DiCoDe method in Algorithm 2, which combines the soft co-design distribution, projected universal guidance and critic distillation into a single framework.

---

**Algorithm 2:** Diffusion Co-Design (DiCoDe)

---

**Input:** memory buffer $\mathcal{D}$, agent $(\boldsymbol{\pi}_\phi, V_\psi)$, diffusion $(\epsilon_\varphi, \mathcal{V}_\vartheta)$
`// Pre-train Diffusion Model`
**for** $i = 1, \ldots, N_{diffusion}$ **do**
    Sample minibatch $\boldsymbol{\theta} \sim u$;
    Train $\epsilon_\varphi$ on $\boldsymbol{\theta}$ with $\mathcal{L}_{\text{DDPM}}$;
`// Agent Training`
**for** $j = 1, \ldots, N_{RL}$ **do**
    Sample batch $\boldsymbol{\theta}$ with PUG$(\epsilon_\varphi, \mathcal{V}_\vartheta)$ as in Algorithm 1 and update $\mathcal{D}$;
    Rollout trajectories in $\theta$ with agent policy $\pi_\phi$;
    Update $(\phi, \psi)$ with MARL algorithm (e.g. MAPPO);
    `// Environment Critic Training`
    **for** $k = 1, \ldots, N_{distill}$ **do**
        Sample minibatch $\boldsymbol{\theta}' \sim \mathcal{D}$;
        Update $\vartheta$ with $\mathcal{L}_{\text{distill}}(\vartheta, \boldsymbol{\theta}')$;

---

In contrast to Gao & Prorok (2023), DiCoDe does not *alternate* between training the environment generator and agent policies. Instead, the same trajectories are used to update both the agent and environment critic, improving sample efficiency. Furthermore, distillation of the agent critic to the environment critic induces knowledge sharing between the two components. Analogous to the warmup phase in off-policy RL, DiCoDe can optionally start with a warmup delay before training the environment critic when environments are sampled from $u$ to prevent overfitting. Alternatively, it is sometimes helpful to add linear annealing to the guidance weighting $\omega$ — a wide coverage of $\Theta$ prevents overfitting. Finally, we optionally choose to run multiple trajectories ($N_{\text{EnvRepeat}}$) on an environment before generating a new batch; this is helpful in simulation when environment generation takes a significant amount of time compared to parallelised rollouts.

## 5 EXPERIMENTAL EVALUATION

In this section, we empirically evaluate the effectiveness of the DiCoDe framework in co-design scenarios. We conduct nine random seeds for each training run and report the mean episode reward. Due to space constraints, we leave the discussion of implementation details to the Appendix A.5.

**Baselines**: Apart from **DiCoDe**, our proposed method, we evaluate against a representative set of baselines[1] and ablations. **RL** refers to the approach by Gao & Prorok (2023), which trains the environment generator with policy gradient. **Fixed** refers to the setting without co-design where the environment is fixed to a sample from $u$, and **DR** refers to domain randomisation (Tobin et al., 2017) where environments are continuously sampled from $u$. **DiCoDe-{Descent, Sampling, ADD, MC}** refer to ablations where (a) we use gradient descent in place of PUG, (b) replace PUG with a top-$k$ sampler, (c) replace PUG with the diffusion guidance method used by Chung et al. (2024), and (d) train the environment critic directly on past trajectory returns instead of targets constructed with distillation. We choose MAPPO (Yu et al., 2022) as the MARL algorithm in our implementation.

**Scenarios**: We evaluate the co-design setting on three challenging tasks (Figure 3). First, we evaluate with D-RWARE, a designable adaptation of the the RWARE (Papoudakis et al., 2021) warehouse management benchmark where robots must collect and deliver packages in a grid world. Then, we assess performance on the WFCRL (Bizon Monroc et al., 2024) windfarm control benchmark to strategise turbine placement with yaw control. Finally, we test with VMAS (Bettini et al., 2022) as a proof-of-concept for multi-agent pathfinding. These three settings cover a diverse set of real-world challenges and are widely used MARL benchmarks. In contrast, prior co-design methods typically restrict their scope to a single class of scenarios.

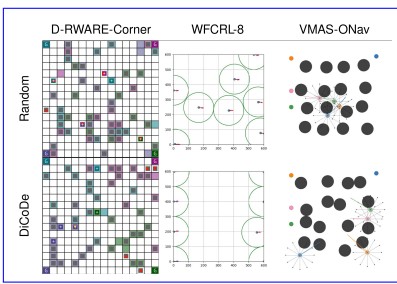

Figure 3: Rendering of environments before and after training.

**1) Performance of DiCoDe relative to prior methods.** In the scenario denoted **Corner**, agents cycle packages between goals located in the four corners and fifty shelves in the gridworld. The goals and shelves are evenly split into 4 colours, and deliveries are constrained to match the colours of goals and shelves. Each method was trained for 20 environment interactions with episodes of 500 interactions each apart from RL which was trained for 60 million interactions. We consider two representations of $\Theta$ (See Appendix A.5.1), where the standard representation is a binary mask of shelves (DiCoDe, DiCoDe-Sampling, DiCoDe-MC) and the alternative representation $\Theta_{Coord}$ is a list of shelf coordinates (DiCoDe-$\Theta_{Coord}$, DiCoDe-Descent).

Training curves for Corner can be seen in Figure 2, left, and we provide quantitative summaries for all experiments in Table 1. These results show that DiCoDe improve multi-agent system performance considerably, converging on successful environment policy pairs with higher rewards than baselines and ablations. In particular, DiCoDe outperforms training on a fixed environment by 26%, demonstrating the tangible benefits of considering the environment as a decision variable. Furthermore, we highlight that DiCoDe delivers 39% more boxes on average than the RL method when measuring performance by a fixed number of policy updates with 66% fewer samples, and 95% more when normalized to the number of samples.

Figure 2, right, visualises the distribution of $\theta$ generated post-training. DiCoDe captures the intuition that shelves should be close to goals of the same color. Furthermore, borders are left clear, possibly as navigation channels. Although DiCoDe-$\Theta_{Coord}$ achieves quantitively similar rewards as the standard representation, the heat-map generated is sharper. We speculate this is related to the interpretation of gradients in the encoding of shelves. Coordinate encodings support small adaptations by moving in the direction of the critic gradient, but in the shelf mask encoding, a small step in the gradient direction leaves the manifold of valid environments. The environments generated lack rigid structure to the human eye, yet achieve impressive performance, suggesting co-design may help explore range of environments otherwise not considered by human experts.

---

[1]We additionally implemented an evolutionary method inspired by ACCEL (Parker-Holder et al., 2022), but could not demonstrate performance above random sampling.

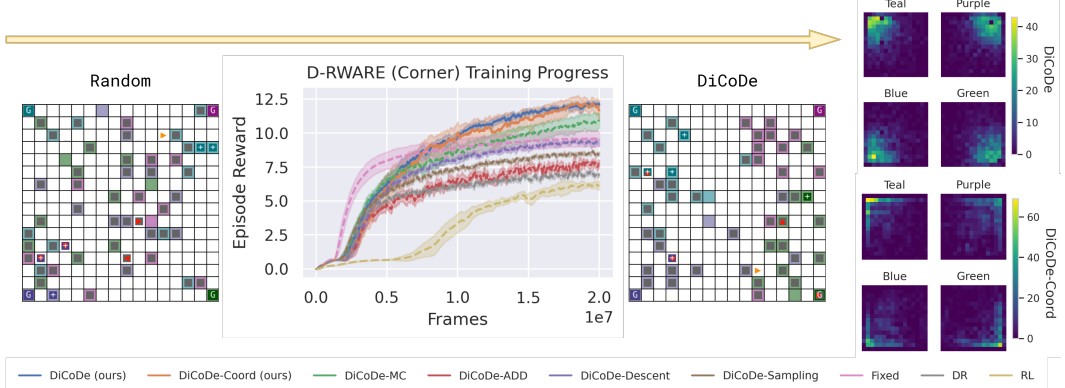

Figure 2: Left) Corner scenario training curves with example of randomly sampled environment and a DiCoDe generated environment after training. We report the mean episode return, smoothed, with 95% confidence intervals shaded. Episode reward corresponds to boxes delivered. Right) Heatmap of shelf placement by DiCoDe across 100 environments. DiCoDe learns to generate from random environments to placing shelves near goals of the same colour with navigation channels free.

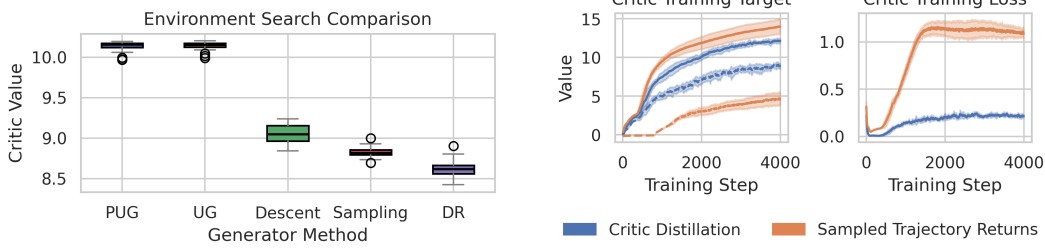

Figure 3: Corner. Left) For each method, we sample 32 environments with guidance from the same critic, and report the value estimated by that critic. Right) Probes of environment critic training. We compare min, max $y$, the learning objective of the environment critic, within each batch generated by DiCoDe (critic distillation) and DiCoDe-MC (sampled trajectory returns). Both are estimates of the true discounted return of an environment. We report the environment critic learning loss.

**2) Ablation on the impact of PUG and Critic Distillation.** Ablations DiCoDe-{Descent, Sampling, ADD} validate the value gain of PUG and DiCoDe-MC validates the value gain of environment critic distillation. The combined DiCoDe method outperforms DiCoDe-Descent by 26%, DiCoDe-Sampling by 48%, DiCoDe-ADD by 57% and DiCoDe-MC by 11%, showing the impact of our contributed modules. We investigate further in two directions.

First, we compare different Sampling, Descent and Universal Guidance (UG) (Bansal et al., 2023) methods compared to PUG, using the same fixed pre-trained environment critic on $\Theta_{\text{Coord}}$. In Figure 3, we observe that PUG and UG obtain similar values exceeding the other methods. This indicates carefully-designed diffusion is an effective search method over $\Theta$: constraint-projection leads to minimal loss in optimisation performance, noting that UG generates invalid environments. The highest-value achieved by sampling the best of 1024 uniformly sampled environments is 12% worse in comparison to the mean of PUG, suggesting future environment design methods should rely on learnt generators rather than replay (Jiang et al., 2021b). Visualisations (see Appendix 7) verify PUG generates environments with distribution of shelves close to goals of the same colour while leaving clear navigation channels. The baseline methods are in local minima, in particular the colour of shelves which are hard to optimise as switching colours is a large jump in $\Theta_{\text{Coord}}$.

Second, we analyse the environment critic targets $y$ generated by DiCoDe against DiCoDe-MC during a training run. Using $y_{\text{distill}}$ confers several noticeable properties in favour of DiCoDe. Notice in Figure 3 how $y_{\text{distill}}$ has a lower maximum and higher minimum than $y_{\text{mc}}$, supporting the claim that critic-generated targets may filter out stochasticity within rollouts of fixed $\theta$. Extreme values of $y_{\text{MC}}$

Table 1: Expected episode rewards at end of training, 0.95 EMA smoothed over training timesteps with 95% confidence intervals across 9 random seeds. ∗: We report normalised to the a fixed number of policy updates, noting the RL method requires more samples per update at 300% for RWARE, 400% for WFCRL and 250% for ONav.

| Scenario | DiCoDe | | Baselines | | | Ablations | | | |
|---|---|---|---|---|---|---|---|---|---|
| | $\Theta$ | $\Theta_{\text{Coord}}$ | RL* | Fixed | DR | Desc. | Sampl. | ADD | MC |
| Corner | $\mathbf{12.1}_{\pm\mathbf{0.2}}$ | $11.7_{\pm 0.7}$ | $8.7_{\pm 0.4}$ | $9.6_{\pm 0.6}$ | $6.9_{\pm 0.1}$ | $9.3_{\pm 0.3}$ | $8.2_{\pm 0.2}$ | $7.7_{\pm 0.3}$ | $10.9_{\pm 0.5}$ |
| WFCRL2 | $\mathbf{490}_{\pm\mathbf{0}}$ | — | $485_{\pm 5}$ | $442_{\pm 28}$ | $443_{\pm 2}$ | — | — | — | — |
| WFCRL4 | $\mathbf{430}_{\pm\mathbf{2}}$ | — | $404_{\pm 6}$ | $387_{\pm 10}$ | $382_{\pm 0}$ | — | — | — | — |
| WFCRL8 | $\mathbf{370}_{\pm\mathbf{5}}$ | — | $323_{\pm 3}$ | $325_{\pm 8}$ | $314_{\pm 1}$ | — | — | — | — |
| ONav | $\mathbf{2.29}_{\pm\mathbf{0.08}}$ | — | $1.92_{\pm 0.09}$ | $2.24_{\pm 0.07}$ | $1.80_{\pm 0.01}$ | — | — | — | — |

may reflect luck rather than true environment quality. Additionally, up until approximately step 800, $y_{\text{mc}}$ remains below 0 due to sampling rollout returns that do not reflect the latest policy. Conversely, $y_{\text{distill}}$ minimum increases earlier, showing mitigation of policy-shift. These results demonstrate critic distillation confers a stable and accurate training signal, improving sample efficiency.

**3) Generalisation to continuous environments and comments on scalability.** We evaluate our method on three windfarm management scenarios, **WFCRL-{2,4,8}**, with the suffix denoting the number of turbines to be placed on a square map. There is a minimum distance constraint between turbines and agents policies control the yaw of each turbine to adjust to wind conditions. Each setup is trained for $903,000$ frames across $6,020$ environments. Additionally, we examine applicability to the multi-agent navigation **VMAS-ONav** scenario, equipped with 16 obstacles that can be reconfigured in their local neighbourhoods. This is trained on $804,000$ frames across $8,040$ environments.

Table 1 shows average returns after training. In these scenarios, the proposed algorithm outperforms baselines by achieving higher episode returns across averaging $9.5\%$ above Fixed environments, $10.3\%$ above RL (despite training on fewer environments) and $17.1\%$ above domain randomisation.

When fine-tuning for WFCRL, we found it essential to anneal the guidance weights in training as discussed in Section 4.4, reflecting PUG enables control over the amount of environment exploration during training. In samples of the windfarms generated by DiCoDe (Figure 3), we see the guided diffusion model learns to split turbines into two groups and distribute them in the major axis of wind to reduce turbulence. These results demonstrate the efficacy of DiCoDe across a wide range of environments, both continuous and discrete, whereas prior methods limit implementation to a single class of scenarios.

In Figure 4, left, we plot the progression of performance as the number of turbines increase which corresponds to increasing number of agents and environment design dimensionality. In contrast to the severe drop-off of RL performance past 4 turbines, DiCoDe maintains performance gains, demonstrating the scalability of our approach. The computational complexity of DiCoDe does not scale with the number of training iterations, taking a constant amount of time each iteration. We do not consider the wall-clock overhead of running diffusion inference significant: the ratio of environment generation cycles relative to the number of samples in an environment in realistic scenarios is negligible.

In Figure 4, right, we visualise representative examples of environments generated by DiCoDe and baselines. In both the ONav and WFCRL4 environments, the DiCoDe-generated examples exhibit structures that lie closer to the boundary of feasible design space, distinguishing them clearly from those produced by the prior state-of-the-art approach. We hypothesize this improvement arises from two factors. First, the sample inefficient Reinforce method may not have fully converged to the optimal solution in the training budget. Second, the diffusion-based generative distribution more effectively captures the multi-modal clusters of performant environments than the multi-variate Gaussian representation employed in Reinforce.

## 6 Discussion

We introduced diffusion co-design (DiCoDe), a novel, state-of-the-art co-design framework for learning highly rewarding policy-environments pairs. DiCoDe incorporates projected universal

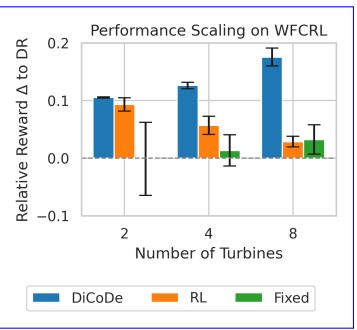 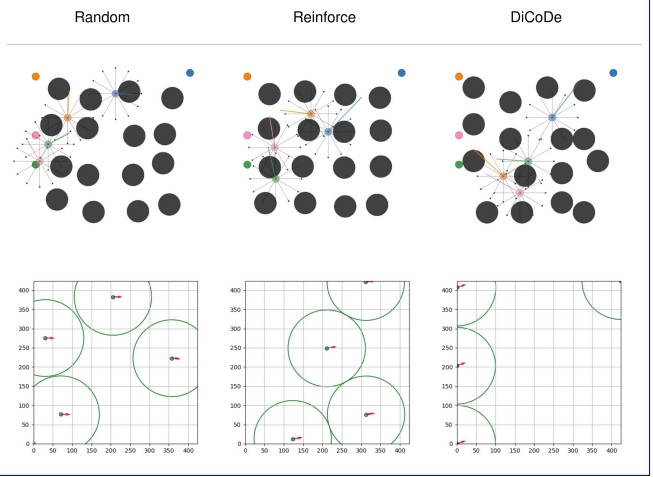

Figure 4: Results on continuous environment design spaces. Left) Performance of co-design methods relative to domain randomisation against the number of turbines in WFCRL. Right) Examples of generated environments after training, with ONav and WFCRL4.

guidance (PUG) for guiding pre-trained diffusion models and critic distillation to improve sample efficiency (by mitigating policy shift and incorporating knowledge of individual agent interactions), and coordinates these techniques with multi-agent reinforcement learning. In empirical evaluations across five scenarios encompassing warehouse delivery, windfarm management and multi-agent navigation, DiCoDe achieves in expectation $16.1\%$ reward above state-of-the-art, and $12.8\%$ above the case without co-design. Collectively, these improvements redefine the limits of multi-agent environment co-design to previously intractable domains.

There exist several directions for future work. Although our method uses an uninformative prior $u$, there is an opportunity to exploit a different underlying distribution by incorporating foundational models (Lehman et al., 2023; Xian et al., 2023) trained on existing datasets of expert-designed environments. Secondly, DiCoDe relies on the soft co-design distribution to explore the environment design space. This can be improved by incorporating unsupervised environment design in a multi-objective framework. Finally, although our method shows strong empirical performance and is built on principled foundations, we do not provide theoretical guarantees. Theoretically examining co-design convergence is of interest.

## 7 REPRODUCIBILITY STATEMENT

We understand the importance of reproducibility, and make efforts to ensure our work is reproducible. We provide detailed explanations of our methodology in Section 4, and discuss the evaluation setup in Section 5 and Appendix A.5. We publicly release our training and evaluation code at ██████████████████████████████████████, which can readily be used to reproduce all results in this paper. We used up-to-date package management practices to enable easy installation of the environment.

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

## A APPENDIX

This section contains additional information on diffusion model background, a comparison of our work with ADD, other use cases of diffusion models in reinforcement learning settings, additional figures of our experiments, and our experiment setup.

## A.1    DENOISING DIFFUSION IMPLICIT MODELS

In this section, we provide additional details on diffusion models, primarily from the perspective of noise addition and removal based on DDPM Ho et al. (2020).

A *forward* diffusion process iteratively adds Gaussian noise to a sample (environment) $x_0$ for $T$ timesteps according to variance schedule $\beta_1, \ldots, \beta_T$ to form a Markov chain.

$$q(x_t|x_{t-1}) = \mathcal{N}\left(x_t; \sqrt{1-\beta_t}x_{t-1}, \beta\mathbf{I}\right)$$
$$q(x_{1:T}|x_0) = \prod_{t=1}^{T} q(x_t|x_{t-1}) \tag{13}$$

Given target distribution $p(x_0)$, the process above defines a series of latent variable distributions $p(x_1), \ldots, p(x_T)$. The distribution of interest is $p(x_0)$ (e.g. a distribution of valid environments), which, although unknown, we may have samples for.

Consider the inverse of the forward process: the *reverse* diffusion process iteratively removes noise until a clean environment remains.

$$p(x_{t-1}|x_t) = \mathcal{N}(x_{t-1}; \mu(x_t, t), \Sigma(x_t, t))$$
$$p(x_{0:T}) = \prod_{t=1}^{T} p(x_{t-1}|x_t) \tag{14}$$

Therefore, learning $p(x_0)$ reduces to matching a reverse process with forward process, using samples from the desired distribution. In our use case, this is the uniform distribution of valid environments.

Ho et al. (2020) introduce DDPM as a concrete method to learn Equation 14. First, assume a linear noise schedule $\beta_t$. We can consider learning a simplified approximation (parameterised by $\varphi$) of the evidence-based lower bound for $p(x_t)$ with surrogate error function $\epsilon_\varphi$ using standard gradient descent techniques. The loss is defined as

$$\alpha_t = \prod_{i=1}^{t} (1 - \beta_i)$$
$$\mathcal{L}_{\text{DDPM}}(\theta) = \mathbb{E}_{t,\epsilon,x_0}\left[||\epsilon - \epsilon_\varphi\left(\sqrt{\alpha_t}x_0 + \sqrt{1-\alpha_t}\epsilon\right)||^2\right] \tag{15}$$

where $\epsilon$ is unit Gaussian noise, $t$ is uniformly sampled between $1, ..., T$ and $x_0$ is a training sample. In other words, $\epsilon_\varphi$ attempts to estimate the time-conditioned noise. It is possible to sample from the target distribution by following the reverse Markov process:

$$\Sigma_\varphi(x_t, t) = \beta_t$$
$$\mu_\varphi(x_t, t) = \frac{1}{\sqrt{1-\beta_t}}\left(x_t - \frac{\beta_t}{\sqrt{1-\alpha_t}}\epsilon_\varphi(x_t, t)\right). \tag{16}$$

In later work, Song et al. (2021a) construct non-Markovian diffusion processes with denoising diffusion implicit models (DDIM) to speed up the reverse sampling process; their method uses the same training procedure as DDPMs. This relies on $\epsilon_\varphi$ as a predictor of $x_0$ as in Equation 9. In our implementation of DiCoDe, we train the diffusion model as in DDPM (Equation 15).

## A.2    UNIVERSAL GUIDANCE AND PROJECTED DIFFUSION MODELS

Recall the score decomposition in Equation 5, which is conditioned on diffusion time $t$. Bansal et al. (2023) introduce *universal guidance* to skip conditioning the classifier on noisy images. Instead, they leverage the information within the expected clean image. Assuming the underlying process is DDIM, *forward guidance* is defined as

$$\hat{\epsilon}_{\varphi,\vartheta}(x_t, t) = \epsilon_\varphi(x_t, t) + \omega\sqrt{1-\alpha_t}\nabla_{x_t}\log c_\vartheta(\hat{x}_0^t|y) \tag{17}$$

where $\omega$ is the guidance strength hyperparameter and $c_\vartheta(\hat{x}_0^t|y)$ is a classifier network. It is possible to use $\hat{\epsilon}_{\varphi,\vartheta}(x_t, t)$ in place of the original estimated noise in the reverse process. In addition to forward guidance, Bansal et al. (2023) introduce *backward guidance* and *recurrence steps*.

Backward guidance improves the conditional guidance bias by replacing the single step gradient, $\nabla_{x_t} \log c_\vartheta(\hat{x}_0^t)$, with the linear interpolation of multiple gradient descent steps, enabling a more accurate direction towards the local minima. In practice, the backward guidance process begins with the result of forward guidance

$$\overline{x}_0^t = \frac{x_t - \sqrt{1-\alpha_t}\hat{\epsilon}_{\varphi,\vartheta}(x_t, t)}{\sqrt{\alpha_t}} \tag{18}$$

and uses the Adam optimiser (Kingma & Ba, 2015) to compute the backward guided prediction as

$$\Delta \overline{x}_0^t = \arg\min_\Delta \log c_\vartheta\left(\overline{x}_0^t + \Delta | y\right)$$

$$\overline{\epsilon}_{\varphi,\vartheta}(x_t, t) = \hat{\epsilon}_{\varphi,\vartheta}(x_t, t) - \sqrt{\frac{\alpha_t}{1-\alpha_t}}\Delta\overline{x}_0^t. \tag{19}$$

Recurrence steps enable inference-time scalingm. For $k$ steps and $x_t^0 = x_t$, iteratively compute

$$x_t^{i+1} = \sqrt{\frac{\alpha_t}{\alpha_{t-1}}}S(x_t^i, \overline{\epsilon}_{\varphi,\vartheta}(x_t^i, t), t) + \sqrt{1 - \frac{\alpha_t}{\alpha_{t-1}}}\mathcal{N}(0, \mathbf{I}) \tag{20}$$

where $S$ is the sampling method of the chosen reverse diffusion process.

Alternative to gradient based guidance, projection methods enforce hard constraints on the generated samples and approximate the constrained score function. For example, post-processing projections (Giannone et al., 2023) can be used on the samples of diffusion models, and Song et al. (2021b; 2022) apply linear projections at each step of the diffusion process to ensure samples are consistent with measurements.

In recent work, Christopher et al. (2024) propose *projected diffusion models* (PDM) as a method to enforce constraints on score diffusion models. Their method may be directly applied to *stochastic gradient Langevin dynamics* (SGLD) (Welling & Teh, 2011) and the sampling method suggested by Song et al. (2021b). At a high level, PDM casts the reverse process as a constrained optimisation problem and theoretically justifies projecting samples onto the constrained domain (assuming a convex constraint set) at each step of the reverse process. However, PDM directly applied to DDPM or DDIM was shown to have poor empirical performance.

## A.3 Comparison to Chung et al. (2024)

DiCoDe is partially inspired by the success of ADD (Chung et al., 2024) in the domain of unsupervised environment design . However, despite structural similarities, there are key methodological differences.

DiCoDCe and ADD share the same pre-training paradigm. Divergence occurs in environment generation and environment critic training. Whereas ADD employs standard classifier guidance, we introduce projected universal guidance. Relative to classifier guidance, PUG is better suited for co-design with its constraint satisfying properties and avoidance of noise-conditioning critics. In our ablations against DiCoDe-ADD (Table 1, PUG is shown to be a key component that leads to improvement in reward. In environment critic training, ADD uses a differentiable regret estimator for the adversarial UED target, while we propose critic distillation in DiCoDe. These two approaches are incomparable due to the different objectives.

## A.4 Diffusion Models in Reinforcement Learning

Diffusion models in reinforcement learning have been utilised in systems beyond ours and Chung et al. (2024). Zhu et al. (2023) Janner et al. (2022) experiment with diffusion models as a trajectory planner for robotic tasks; they leverage classifier guidance and find that physical constraints can be adequately posed as an in-painting problem. Wang et al. (2023), Chen et al. (2023), Chi et al. (2023) andRen et al. use diffusion models as an expressive policy class with success in multi-modal action and trajectory distributions. Sayar et al. (2024) use diffusion models as a goal-distribution generator for curriculum learning. Concurrently with our work, recently Ghosh et al. (2025) developed

Figure 5: D-RWARE: Robots (orange triangles) are rewarded for bringing requested boxes (+) from shelves (shaded grids) to goals (G). Goals and boxes should be the same colour, and empty boxes should be placed back onto shelves.

a diffusion-based hardware accelerator generator to replace reinforcement learnign and sampling techniques.

## A.5 EXPERIMENTAL DETAILS

We discuss the experimental setup, including scenarios, hyperparameters and compute required.

### A.5.1 SCENARIOS

**Designable Multi-Agent Warehouse.** Warehouse layout design and application is an important real-world problem, accounting for above 30% of logistic costs (Roodbergen et al., 2015), inciting significant research interest: the recent survey by Albert et al. (2023) reviewed 3798 papers over a 20-year timeframe. Multi-robot warehouse (RWARE) (Papoudakis et al., 2021) is a widely used MARL benchmark inspired by real-world warehouse management tasks. In RWARE, a team of robots collaboratively pick up (uniformly sampled) requested boxes from shelves and deliver them to goals — a reward is received each time a box is delivered, and empty boxes must be returned to shelves. Shelves act as obstacles, interfering with agent navigation — *a designer must strike a careful balance between placing shelves close to goals and freeing movement channels*.

As part of our contributions, we fork RWARE and propose a new environment Designable Multi-Robot Warehouse (D-RWARE), shown in Figure 5. D-RWARE extends RWARE with a number of improvements including an environment design API, coloured objectives, and reward shaping. The D-RWARE scenario is a configurable grid world with a fixed number of robots, shelves and goals. Agents interact with the world using a discrete *action space*: movement in the four cardinal directions and picking/dropping boxes on their square. They receive *observations* on (shelves, boxes and teammates) within a certain distance from the agent, heuristics to the nearest (requested box, goal and empty shelf), and personal status information. We select a convolutional neural network (CNN) (LeCun et al., 1989) architecture, followed by an MLP head for the policy, and share parameters between different agents.

In RWARE, an agent receives a reward of +1 for each box delivered to a goal. We apply a shaped reward in D-RWARE to reduce the sparseness of the reward signal: part of the reward allocation is transferred to picking up a requested box, bringing requested boxes closer to goals, and returning empty boxes to shelves. Because the reward shaping is potential-based[2] Ng et al. (1999), we can keep the original interpretation of episode returns as the count of boxes delivered.

The agent critic should be able to evaluate the expected return of a policy across different environments. We use the same UNET encoder with attention (Ronneberger et al., 2015; Ho et al., 2020) as ADD for the backbone of our critic network. The agent critic network takes in agent observations, concatenated with a global map of the environment, to estimate the expected return of the agent policy.

---

[2]We ignore the discount factor in shaping for simplicity, so there may be minor changes to the optimal policy.

A key design choice of DiCoDe is selecting suitable $\Theta$, $\boldsymbol{X}$, and $\mathfrak{P}_\Theta$ for the diffusion process; the representation can implicitly encode invariances and structural constraints. We assume goal and agent positions are known in advance, and examine two possible representations for deciding the layout of shelves.

- **Standard**: The layout of shelves is represented as a binary mask for each colour, where each pixel represents a square in the grid world. Let $\mathbb{Z}_N^+$ denote $\{1, 2, \ldots, N\}$ and $C = \mathbb{Z}_{N_{\text{colours}}}^+$ is the set of colours

$$\boldsymbol{X}_{\text{image}} = \mathbb{R}^{H \times W \times N_{\text{colours}}}$$

$$\Theta_{\text{image}} = \{\theta \in \boldsymbol{X} : \theta_{i,j,c} = 1 \text{ if square } (i,j) \text{ has shelf of colour } c, \text{ else } 0\}$$

  This representation assigns each shelf to a single square in the grid world, and the natural CNN architecture choice is invariant to translations, which aids neural network training. Although $\epsilon_\varphi$ adequately guides boxes of different channels to different squares and pushes real values to binary, it insufficiently constrains the number of shelves within a channel. Therefore, projection operation $\mathfrak{P}_{\Theta_{\text{image}}}$ sorts the pixels in a channel by value, retains the specified number (shelves) of top-ranked values, followed by transformation to a binary mask. A UNET is suitable for the diffusion model $\epsilon_\varphi$, and we use the same UNET encoder architecture as the agent critic for the environment critic $\mathcal{V}_\vartheta$.

- **Coord**: Alternatively, we can represent shelves as a set of coordinate-colour pairs.

$$\text{Shelf}^{\boldsymbol{X}} = \mathbb{R} \times \mathbb{R} \times C$$

$$\boldsymbol{X}_{\text{Coord}} = \{\text{Shelf}_1^{\boldsymbol{X}}, \ldots, \text{Shelf}_{N_{\text{shelves}}}^{\boldsymbol{X}}\}$$

$$\text{Shelf}^\Theta = \mathbb{Z}_{\text{Width}}^+ \times \mathbb{Z}_{\text{Length}}^+ \times C$$

$$\Theta_{\text{Coord}} = \{\text{Shelf}_1^\Theta, \ldots, \text{Shelf}_{N_{\text{shelves}}}^\Theta\}$$

  $\boldsymbol{X}_{\text{Coord}}$ will constrain the correct number of shelves but does not snap locations to $\Theta_{\text{Coord}}$. We use the Hungarian algorithm (Kuhn, 1955) to match shelves to the closest grid squares, where the cost function is the Manhattan distance between the shelf and the grid coordinate. Then, we move shelf coordinates linearly towards the matched grid square until the target grid square is the closest grid square for $\mathfrak{P}_{\Theta_{\text{Coord}}}$. At the end of the diffusion process, we snap shelf coordinates exactly — the prior projections guarantee this will lead to a valid environment.

  Empirically, an MLP suffices for $\epsilon_\varphi$. To select a suitable architecture for the environment critic model, we evaluate[3] a UNET decoder (as in the image representation) preceded by a graph attentional layer Veličković et al. (2018): the graph attentional layer takes in the shelf coordinates as nodes, and connects edges (encoded with radial distance) from shelves to nearby grid points. Initial node encodings for shelves are one-hot encodings of the shelf colour. The grid points, after the graph attentional layer, can then be interpreted as pixels in an image by the CNN. By construction, this architecture is invariant to the permutation of shelves and also captures the spatial relationships between shelves and grid points. In the limiting case where shelf coordinates are perfectly aligned to the grid, the architecture is equivalent to $\Theta_{\text{image}}$ representation.

**Wind Farm Control (WFCRL).** The increasing demand for clean energy is leading to rising industrial and academic interest in designing efficient wind farms (Wang et al., 2015; Hou et al., 2019). The primary objective of wind farm control lies in minimising wind power losses by the wake interaction (Jensen, 1983) caused by turbulence from upstream turbines; a secondary objective may be to *reduce mechanical fatigue*. Both the control policy and farm layout have a direct impact on this objective. To provide a tool to aid the development of agent-based wind farm control policies, Bizon Monroc et al. (2024) introduce Wind Farm Control with Reinforcement Learning (WFCRL), an open-source MARL environment for the wind farm control problem with adjustable layouts.

A WFCRL scenario consists of 10 homogenous turbine agents spread out on a $W \times H$ map with a minimum distance constraint Kusiak & Song (2010) between turbines. Agents receive local measurements of the wind conditions (speed and direction) as observation, concatenated with the layout.

---

[3]We also experiment with E(n) equivariant neural networks (Satorras et al., 2021), with unsatisfactory performance.

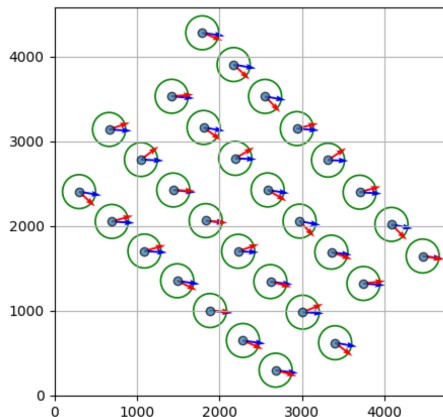

Figure 6: WFCRL: Wind farm layout representing the Ormonde offshore wind farm Abritta (2023). Circles represent individual turbines, and the green border constrains the minimum distance between turbines. Blue arrows show wind direction, and red arrows show turbine yaw. In the real world, wind farms often place turbines in a grid layout.

Using these observations, agents may adjust their yaw to balance between maximising local power product and deflecting wake away from downstream turbines. The team of turbines recieve the same reward as the mean power subtracted by fatigue. The scenario's transition function depends on an underlying wind condition simulator; we choose the FLORIS (Gebraad et al., 2016) simulator option and sample initial free wind conditions from the Weibull distribution.

In our implementation of DiCoDe for WFCRL, we parametrise the diffusion model $\epsilon_\varphi$ with an MLP. We assume there are available communication links between the turbines: the policy $\pi$ is parame- terised by an E(3)[4] equivariant graph neural network (GNN) (Satorras et al., 2021). To transform the set of turbines into a graph, we build a fully connected structure with attention weightings on edges. Similarly, we parametrise the agent critic with E(3) invariant GNN — an equivariant GNN followed by an invariant aggregation layer. The environment critic takes in turbine positions as input without wind directions. Therefore, we use a translationally invariant GNN that is not invariant in rotations and reflections.

To enforce the minimum distance constraint, we formulate $\mathfrak{P}_{\Theta_{\text{wfcrl}}}$ as a soft constraint to penalise constraint violations while trying to minimise movement of turbine locations; this is solved with gradient descent. To enforce hard constraint satisfaction, we apply a Sequential Least SQuares Programming (SLSQP) solver (Schittkowski, 1983) to the final layout.

**Multi-agent navigation** is a mandatory subroutine in robotic application settings such as ware- houses, factories, or hospitality. Additionally, it is the setting considered in prior work for compar- ison (Gao & Prorok, 2023). We implement a multi-agent navigation scenario as a using the VMAS (Bettini et al., 2022) multi-agent physics simulator. In our formulation, each agent is spawned in a fixed position, and is rewarded for approaching a fixed goal. We parametrise the diffusion model, agent policy, agent critic and environment critic with MLPs, and set up the obstacles with a local boundary such that constraints are not necessary in the environment. We remark that because both agent critic and environment critic use the same information processing architecture, and that the environment setup is differentiable, this is an edge case of distillation where agent critic and envi- ronment critic may share parameter weights. Finally, we note that the training time on multi-agent navigation with our hyperparameter selection is an order of magnitude lower than D-RWARE (20 minutes compared to 30 hours, and that prior co-design works were often limited to only multi-agent navigation problems.

---

[4]Group of rotations, reflections and translations in 3D.

## A.5.2 HYPER-PARAMETERS

We use the MAPPO implementation of TorchRL (Bou et al., 2024) and our diffusion pipeline is forked from Chung et al. (2024), which itself is a fork of Yoon et al. (2023).

| MAPPO HP | Value | | |
|---|---|---|---|
| | D-RWARE | WFCRL | VMAS |
| Optimiser | | Adam | |
| Learning rate annealing | | Cosine (Restartless) | |
| Initial actor LR | | 3e-4 | |
| Final actor LR | | 0 | |
| Initial critic LR | | 3e-4 | |
| Discount factor ($\gamma$) | | 0.99 | |
| Clip ratio ($\epsilon$) | | 0.2 | |
| Max gradient norm | | 1.0 | |
| Critic loss criterion | | Huber | |
| Final critic LR | 1e-4 | 2e-4 | 1e-4 |
| GAE parameter ($\lambda$) | 0.9 | 0.95 | 0.9 |
| Entropy coefficient | 1e-3 | 0 | 1e-3 |
| Update epochs | 5 | 8 | 10 |
| Minibatch size $M$ | 500 | 150 | 400 |
| Minibatches per epoch | 10 | 20 | 10 |
| Normalise advantage | False | True | False |
| Critic normalisation | False | True | False |

Table 2: MAPPO Hyperparameters used in experiments on Corner, Rect-8 and Square-10. Critic normalisation refers to an adaptation of C66 in Andrychowicz et al. (2021) where instead of running averages we pre-compute the mean and std used by running a heuristic policy.

Table 2 lists the MAPPO hyper-parameters used in experiments. Table 3 lists additional hyperparameters.

| DiCoDe HP | Value | | | |
|---|---|---|---|---|
| | D-RWARE $\Theta$ | D-RWARE $\Theta_{\text{Coord}}$ | WFCRL | VMAS |
| Diffusion Steps | | 1000 | | |
| Diffusion Process | | DDIM (50 steps) | | |
| Optimiser | | Adam | | – |
| $\mathcal{D}$ buffer size | | 8096 | | – |
| Warmup Environment # | | 2048 | 400 | 400 |
| LR | | 3e-5 | 1e-4 | – |
| $N_{\text{EnvRepeat}}$ | | 10 | 1 | 1 |
| Loss Criterion | | MSE | Huber | – |
| Batch size | | 64 | 32 | – |
| $M_{\text{distill}}$ | | 3 | 3 | – |
| Recurrences | | 8 | 4 | 8 |
| $\omega$ | 200 | 5 | $0 \rightarrow 3$ | 50 |
| Backward | 0 | 16 (LR=0.01) | 0 | 6 (LR=0.01) |

Table 3: DiCoDe Hyperparameters used in experiments. The environment critic in VMAS is not trained, but updated with the latest agent critic weights.

For other hyper-parameters not listed, please refer to the codebase with yaml configuration files.

### A.5.3 TRAINING HARDWARE

Experiments were run on several different devices.

The first device had a single NVIDIA RTX 3090 GPU with 24GB of VRAM. The device used an Intel i5-13600KF CPU with 14 cores and 64GB of RAM, running Endeavour OS.

The second device had a single NVIDA RTX 4090 GPU with 24GB of VRAM. The device used an AMD Ryzen 7 7800X3D CPU with 8 cores and 64GB of RAM, running Windows 11 Pro and WSL.

The third device had a single NVIDIA RTX 5090 GPU with 32GB of VRAM. The device used an AMD Ryzen 9 9950 CPU with 16 cores and 64GB of RAM, running Endeavour OS.

The first server has 4 NVIDIA RTX2080TI GPUs, each with 12GB of VRAM. The device used an Intel Xeon Gold 6248R CPU with 48 cores, running Ubuntu 22.04. Experiments were run with Docker.

The second server has 4 NVIDIA L40S GPUs, each with 48GB of VRAM. The device used an Intel Xeon Platinum 8452Y CPU with 72 cores, running Ubuntu 22.04. Experiments were run with Docker.

This work was performed using HPC resources, which we will disclose after the reviewing period.

### A.6 ADDITIONAL RESULTS

We visualise the environments generated by our ablations in Figure 7 for qualitative analysis. The results reveal clear, intuitive structures present in the PUG example, where navigation channels — a space of at least one cell — are present, and colours cluster together. In contrast, the examples obtained through Descent or Sampling are in local minima, particularly the colours of shelves. Additionally, the Sampling method exhibits an untraversable goal in the bottom left corner.

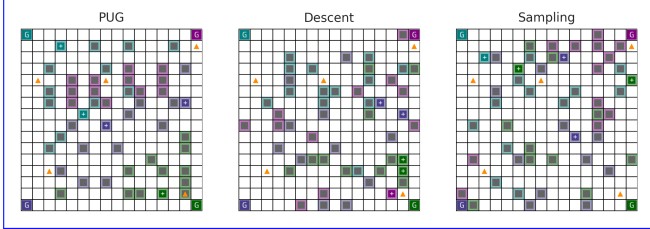

Figure 7: Examples of environments generated using the same critic with projected universal guidance, gradient descent and best-out-of-$k$ sampling.

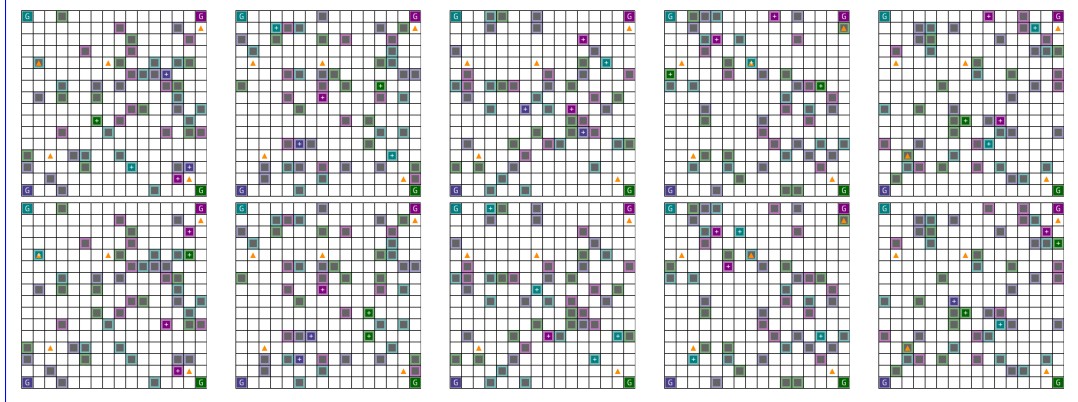

Figure 8: Examples of environments generated at the start of training following a uniform distribution, RWARE Corner.

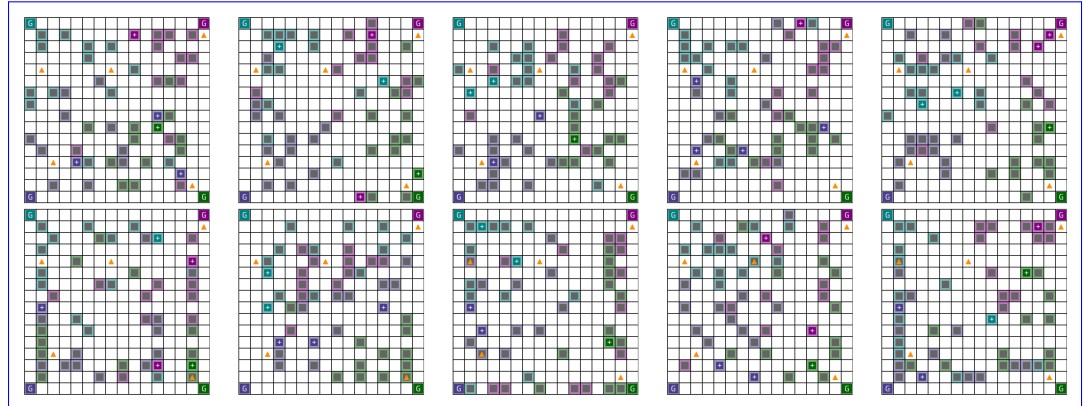

Figure 9: Examples of environments generated at the of training using DiCoDe, RWARE Corner. The top row corresponds to environments sampled in the image diffusion domain, and the bottom the coordinate domain.

We provide additional examples of generated environments in the D-RWARE Corners environment, at the start and end of training, in Figures 8 and 9.

### A.7 LLM DISCLOSURE

We use LLM generated output for word/phrasing suggestions in writing, and error-checking. We also use co-pilot for line-level code auto-completion, and to assist in figure generation (with data processing written by hand).

### A.8 SOFTWARE DEPENDENCIES

We use `uv` for our package management. Table 4 shows the core dependencies used in this project.

| Package Name | License |
|---|---|
| matplotlib | PSF |
| numpy | BSD |
| rware | MIT |
| ADD | CC BY-NC 4.0 |
| torchrl | MIT |
| torch | BSD |
| wandb | MIT |
| hydra-core | MIT |
| pydantic | MIT |
| torch-geometric | MIT |
| torch-scatter | MIT |
| wfcrl | Apache 2.0 |
| seaborn | BSD |
| scipy | BSD |
| uv | Apache 2.0 |
| vmas | GPL-3.0 |

Table 4: Software Dependencies with Licenses

