# OpenReview forum: "Scaling Multi-Agent Environment Co-Design with Diffusion Models"
_ICLR.cc/2026/Conference — Submitted to ICLR 2026_

### Official Review · Reviewer_rKbS · 2025-10-23

**Soundness:** 2
**Presentation:** 2
**Contribution:** 2
**Rating:** 2
**Confidence:** 5

**Summary:**

The paper proposes **Diffusion Co-Design (DiCoDe)**, a framework that jointly optimizes **multi-agent policies** and **environment configurations** using **guided diffusion models**. The goal is to improve **sample efficiency** and **scalability** in high-dimensional environment design spaces. The method introduces two key components:

1. **Projected Universal Guidance (PUG)**: A sampling technique that guides diffusion models to generate **high-reward environments** while satisfying **hard constraints** (e.g., obstacle separation).
2. **Critic Distillation**: A mechanism that transfers knowledge from the **agent critic** to an **environment critic**, providing a **dense and up-to-date learning signal** for the diffusion model.

The method is evaluated on three domains: **warehouse automation (D-RWARE)**, **wind farm control (WFCRL)**, and **multi-agent pathfinding (VMAS)**. Results show **up to 39% higher reward** and **66% fewer simulation samples** compared to prior co-design methods.

**Strengths:**

- **Originality**:
  DiCoDe is the **first method to apply guided diffusion models** to **multi-agent environment co-design**, and it introduces **PUG**, a novel constraint-aware sampling technique that combines **universal guidance** with **projected constraints**.

- **Quality**:
  The paper provides **strong empirical results** across **three diverse domains**, showing **consistent improvements** over baselines. The **critic distillation** mechanism is well-motivated and addresses **policy shift**, a known issue in co-design.

- **Clarity**:
  The method is described **systematically**, with **clear algorithms**, **pseudocode**, and **ablations**. The **visualizations** (e.g., shelf placement heatmaps) help illustrate the **learned environment structures**.

- **Significance**:
  If scalable, DiCoDe could **fundamentally improve** how we design **real-world multi-agent systems** (e.g., warehouses, wind farms), where **environment layout** and **agent policy** are **tightly coupled**.

**Weaknesses:**

### W1. **Scalability vs. Algorithm Design is Unclear**
- While the paper claims **scalability**, the **computational cost** of **guided diffusion** **increases** with the **number of agents** and **environment dimensionality**.
- **No complexity analysis** is provided for **PUG sampling** or **critic distillation** w.r.t. **agent count**.
- **Missing experiment**: **Scaling curves** with **increasing agents** (e.g., 4→16→32) to **quantify** how **wall-clock time** or **memory** grows.

### W2. **Limited Benchmarking on Standard MARL Tasks**
- All experiments are on **custom or modified environments** (D-RWARE, WFCRL, VMAS).
- **No evaluation** on **widely-used MARL benchmarks** like:
  - **SMAC** (StarCraft Multi-Agent Challenge)
  - **MPE** (Multi-Agent Particle Environments)
  - **SMACv2** (stochastic version)
- This **limits generalizability**—it’s **unclear** whether DiCoDe **outperforms SOTA methods** in **standard cooperative/competitive settings**.

### W3. **Cooperation Mechanism is Under-Explained**
- The paper **asserts** that DiCoDe **improves cooperation**, but **does not explain** **how** or **why**.
- **No ablation** on **joint vs. individual rewards**, **communication**, or **emergent specialization**.
- **No visualization** of **agent behaviors** (e.g., trajectories, role separation) to **support** the **cooperation claim**.
- **Missing metric**: **cooperation-specific measures** (e.g., **joint action diversity**, **team coherence**, **inter-agent distances**) are **not reported**.


While DiCoDe introduces **novel ideas** and shows **strong results** in **custom domains**, the **lack of standard benchmark evaluation**, **unclear scalability analysis**, and **vague cooperation claims** **limit its contribution**. The **core novelty** (PUG + critic distillation) is **interesting**, but **without broader validation**, it **remains unclear** whether the method **generalizes** beyond **specific, engineered environments**. I encourage the authors to:

1. Evaluate on **SMAC/MPE** benchmarks.
2. Provide **scaling curves** with **agent count**.
3. Clarify **how cooperation is improved** with **quantitative evidence**.

**Questions:**

### Q1. Scalability Analysis
Can you provide **wall-clock time** and **GPU memory usage** as a function of **agent count** (e.g., 4, 8, 16, 32)?
How does **PUG sampling time** scale with **environment dimensionality**?

### Q2. Standard Benchmark Evaluation
Why not evaluate on **SMAC** or **MPE**?
Can you run **DiCoDe vs. QMIX/MAPPO** on **3s5z** or **simple_spread** to **validate generalizability**?

### Q3. Cooperation Mechanism
What **specific behaviors** emerge that indicate **better cooperation**?
Can you provide **trajectory visualizations** or **role specialization plots**?
Did you measure **joint action entropy**, **inter-agent distances**, or **task allocation patterns**?

### Q4. Ablation on Critic Distillation
What happens if you **remove critic distillation** and **train the environment critic only on returns**?
How **sensitive** is the method to **M_distill** (number of Monte-Carlo samples)?

### Q5. Constraint Handling in PUG
How does **PUG** perform when **constraints are non-convex** (e.g., **connectivity**, **visibility**, **dynamic obstacles**)?
Can you show a **failure case** where **PUG violates constraints** or **gets stuck in local minima**?

---

> ### Author Response · Authors · 2025-11-21
>
> We appreciate the feedback provided. However, we would like to take this opportunity to address some misunderstandings in the evaluation of our work. We clarify that DiCoDe is a scalable and sample-efficient framework for jointly optimising agent and **environment parameters**, and is orthogonal to other developments in the typical multi-agent reinforcement learning setting.
>
> **Q1 and W1: The computational cost of guided diffusion increases with the number of agents and environment dimensionality.
> No complexity analysis is provided for PUG sampling or critic distillation w.r.t. agent count. Missing experiment of scaling with increasing agents (e.g., 4→16→32) to quantify how wall-clock time or memory grows.**
>
> While any non-constant algorithm would result in an increasing computation cost with dimensionality, our experiments with DiCoDe show we are able to consistently generate environments at a larger scale than the prior state-of-the-art (RL [Gao et al.]). Table 1 provides a quantitative comparison of these methods, and we explicitly discuss scaling in the second-to-last paragraph of Section 5.
>
> We emphasise our work is concerned with increasing environment dimensionality, not the number of agents. The cost of DiCoDe training increases with the number of agents, equivalent to the underlying MARL algorithm. That said, in Figure 4, we plot the progression of performance as the number of wind turbines increases. This is simultaneously increasing both the agent number and environment dimension. DiCoDe maintains performance gains in contrast to the dropoff of RL.
>
> The complexity of general diffusion models and projection operations depends on many hyperparameters such as the domain-specific environment representation, model architecture, constraints, and the projection operator chosen. This is out of the scope of this paper, and we refer to the literature on improving inference complexity or parallelisation (Chen et al.) for diffusion models. Notably, we did not increase the number of DDIM steps with environment dimensionality in our experiments.
>
> Finally, the ratio of diffusion inference to environment sampling is minimal, on the order of one environment to multiple episodes and thousands of samples. Even though the wall-clock time of running PUG is significant in experimental settings, we believe this does not reflect eventual real-world application where the time cost of environment sampling can be orders of magnitudes higher (Gabriel et al.).
>
> * Gao, Z., & Prorok, A. (2023). Constrained Environment Optimization for Prioritized Multi-Agent Navigation. IEEE Open Journal of Control Systems, 2, 337-355.
> * Chen, H., Ren, Y., Ying, L., & Rotskoff, G. (2024). Accelerating Diffusion Models with Parallel Sampling: Inference at Sub-Linear Time Complexity. In Advances in Neural Information Processing Systems (pp. 133661–133709).
> * Gabriel Dulac-Arnold, Daniel Mankowitz, & Todd Hester. (2019). Challenges of Real-World Reinforcement Learning.

---

> ### Author Response · Authors · 2025-11-21
>
> **Q2 and W2: Experiments are on custom or modified environments (D-RWARE, WFCRL, VMAS) with no evaluation on widely used MARL benchmarks like SMAC, MPE, SMACv2. This limits generalisability.**
>
> We respectfully disagree with the characterization of our benchmarking. Our work proposes a framework that jointly optimises multi-agent policies and **environment configurations**. Although the three benchmarks suggested (SMAC, MPE, SMACv2)  are  standard for MARL policy learning, they are irrelevant to our setting as they feature fixed environments and **do not incorporate an element of environment design.**
>
> We demonstrate that our approach is generalisable by selecting well-known benchmarks in MARL applicable to agent-environment co-design.
> - RWARE [Christianos et al.] (394 stars on Github). We develop D-RWARE to provide an API for environment design, and add colors to emphasise the impact of the environment.
> - VMAS [ Bettini et al.] (483 stars on Github) and is a state-of-the-art benchmark for MARL, and contains all scenarios within MPE. VMAS is widely used, and policies trained have been transferred to real-world robots (Blumenkamp et al.).
> - WFCRL [Bizon et al.]  is a well-known, standard benchmark for windfarm wake control, a realistic application of environment co-design.
>
> Note prior co-design methods mostly evaluate within a single set of environments. For example, [Gao et al]. solely consider multi-agent navigation, which we replicate with VMAS, and [Roodenberg et al]. consider warehouse layout, a domain we evaluate using D-RWARE.
>
> The environments we select are most suitable to the multi-agent co-design domain, and our results demonstrate DiCoDe generalises across a wide range of both discrete and continuous environments.
>
> * Blumenkamp, J., Shankar, A., Bettini, M., Bird, J., & Prorok, A. (2024). The cambridge robomaster: An agile multi-robot research platform. In International Symposium on Distributed Autonomous Robotic Systems (pp. 439–456).
> * Filippos Christianos, Lukas Sch¨afer, and Stefano Albrecht. Shared experience actor-critic for multi-
> agent reinforcement learning.  Advances in Neural Information Processing Systems, volume 33, pp. 10707–10717.
> * Bettini, M., Kortvelesy, R., Blumenkamp, J., & Prorok, A. (2022). Vmas: A vectorized multi-agent simulator for collective robot learning. In International Symposium on Distributed Autonomous Robotic Systems (pp. 42–56).
> * Bizon Monroc, C., Busic, A., Dubuc, D., & Zhu, J. (2024). WFCRL: A Multi-Agent Reinforcement Learning Benchmark for Wind Farm Control. Advances in Neural Information Processing Systems, 37, 133254–133281.
> * Roodbergen, K., Vis, I., & Taylor Jr, G. (2015). Simultaneous determination of warehouse layout and control policies. International Journal of Production Research, 53(11), 3306–3326.
> * Gao, Z., & Prorok, A. (2023). Constrained Environment Optimization for Prioritized Multi-Agent Navigation. IEEE Open Journal of Control Systems, 2, 337-355.

---

> ### Author Response · Authors · 2025-11-21
>
> **Q3 and W3: DiCoDe and agent-agent cooperation**
>
> Analysing emergent cooperation is indeed a fascinating area of research. However, we wish to clarify DiCoDe’s focus is a method or multi-agent environment co-design in cooperative settings.  Consequently, the most appropriate metric to evaluate our success is the global task reward — a standard measure of performance.
>
> Better environments may lead to improved cooperation as a byproduct of optimising for global task reward. This idea was explored by [Gao et al.] in Section V.d, and is orthogonal to our efforts in scaling with environment dimensionality.  In general, efforts to improve cooperation are largely orthogonal to our method. While we use MAPPO as a representative MARL backbone, this may be replaced with any other MARL algorithm, including those with agent-specialisation (Bettini et al.).
>
> * Bettini, M., Shankar, A., & Prorok, A. (2023). Heterogeneous multi-robot reinforcement learning. arXiv preprint arXiv:2301.07137.
> * Gao, Z., & Prorok, A. (2023). Constrained Environment Optimization for Prioritized Multi-Agent Navigation. IEEE Open Journal of Control Systems, 2, 337-355.
>
> **Q4: Ablation on critic distillation. What happens if you remove critic distillation? How sensitive is the environment to the number of Monte-Carlo samples?**
>
> Critic distillation is a cornerstone of our work and we would like to highlight this exact ablation described in
>
> *Page 8, Section 5.2: Ablation on the impact of PUG and Critic Distillation*
>
> As described in our paper, DiCoDe-MC refers to the ablation where we “train the environment critic directly on past trajectory returns instead of targets constructed with distillation”. DiCoDe confers several advantages over DiCoDe-MC in our experiment:
> - Critic distillation provides a stable learning signal by filtering out stochasticity within rollouts. This leads to a training target with lower maximum and higher minimum.
> - Critic distillation mitigates policy shift, whereas the replay buffer of environment returns leads to inaccurate targets such as negative return rollouts that do not reflect the latest policy.
>
> In D-RWARE, distillation provides an 11% improvement to training on environment returns. Additionally, we find a low $M_{distill}$ is sufficient for strong training results, and set it as 3 (See Appendix A.5.2). We would be happy to clarify any additional questions about the setup of this experiment.
>
> **Q5: Constraint handling in PUG. How does PUG perform when constraints are non-convex? Can you show a failure case violating constraints or local minima?**
>
> Thank you for this question.
>
> **Constraint Violation**: PUG is guaranteed never to violate constraints, provided the projection operator correctly projects to a valid environment. This is because the final step of the PUG sampling process (Algorithm 1) includes a projection onto the feasible set.
>
> **Non-convexity and Local Minima**: While we do not establish theoretical claims for PUG regarding global optimality, we show strong empirical performance on complex design spaces, including the pixel domain for D-RWARE where the set of valid environments is highly non-convex and disconnected (see Figure 2). In Figure 3 (Left), we ablate PUG against other optimization methods (gradient descent and top-k sampling). PUG consistently finds environment samples with significantly higher values, demonstrating its effectiveness as a search method in this context.

---

### Official Review · Reviewer_SXPT · 2025-10-26

**Soundness:** 3
**Presentation:** 3
**Contribution:** 3
**Rating:** 6
**Confidence:** 3

**Summary:**

This paper introduces Diffusion Co-Design (DiCoDe), a novel framework for scalable and sample-efficient multi-agent environment co-design, where both agent policies and environment parameters are jointly optimised to maximise system performance. Existing co-design approaches suffer from combinatorial explosion and instability due to policy–environment non-stationarity. DiCoDe overcomes these issues through two key innovations: (1) Projected Universal Guidance (PUG), a constraint-aware diffusion sampling technique that enables diffusion models to generate valid, high-reward environment configurations while preserving diversity; and (2) Critic Distillation, which transfers dense value estimates from the MARL critic to a separate environment critic that provides low-variance, adaptive guidance to the diffusion process. This design allows the diffusion model to remain fixed after pretraining, while the critic dynamically steers environment generation as agents learn. Evaluated across three benchmarks — warehouse logistics (D-RWARE), wind farm control (WFCRL), and cooperative navigation (VMAS) — DiCoDe achieves up to 39% performance gains and 66% fewer samples compared to existing co-design baselines, demonstrating its scalability and stability in high-dimensional multi-agent design spaces.

**Strengths:**

1. The paper introduces Diffusion Co-Design (DiCoDe), a new framework for _jointly_ optimising multi-agent policies and environment parameters — a direction previously limited by scalability and sample inefficiency.
2. The paper proposes Projected Universal Guidance (PUG) that is a principled sampling technique that merges universal guidance and projected diffusion models, enabling generation of high-reward, constraint-satisfying environments. This addresses the infeasibility of standard classifier guidance and avoids invalid samples. This paper also proposes Critic Distillation that is a mechanism that distils information from the MARL agent’s critic into an environment critic, providing a dense and up-to-date learning signal. This reduces variance, mitigates policy-shift, and vastly improves sample efficiency — a key bottleneck in earlier works.
3. Across three diverse benchmarks — D-RWARE (warehouse logistics), WFCRL (wind-farm control), and VMAS (multi-agent navigation) — DiCoDe consistently surpasses baselines such as MAPPO-based co-design and domain randomisation. In general, DiCoDe achieves 39% higher task rewards and 66% fewer samples in warehouse logistics. In general, DiCoDe maintains strong performance as the number of agents/design parameters scales (e.g., from 2→8 turbines), unlike prior RL-based co-design methods that degrade sharply. More importantly, it demonstrates adaptability of DiCoDe to both discrete and continuous design spaces — a rare accomplishment among environment design algorithms.
4. The paper includes thorough ablation studies verifying the impact of both PUG and critic distillation: (1) Removing PUG or replacing it with prior sampling/guidance methods reduces performance by 26–57%; (2) Omitting critic distillation reduces stability and increases noise, validating its necessity. Qualitative results (e.g., shelf placement heatmaps) visually confirm that DiCoDe learns functionally interpretable structures.
5. This paper has provided implementation details, baselines, and hyperparameters are fully disclosed, with code promised for release. Also, nine-seed averages with confidence intervals indicate credible empirical validation. Evaluation spans multiple environment modalities, not confined to a single task class, strengthening generality claims.
6. This paper moves the co-design field closer to real-world deployment, bridging simulation-based MARL and practical layout/dynamics optimisation. More importantly, it provides a unifying framework that could be extended to unsupervised environment design and multi-objective optimisation, as the authors note in discussion.

**Weaknesses:**

1. The paper lacks a formal theoretical analysis of convergence or stability for the co-design process.  While the diffusion-based sampling and critic distillation are well-motivated empirically, there is no formal justification (e.g., fixed-point or equilibrium guarantees) that the joint optimisation of agents and environment converges to any optimal co-design solution. The method’s stability arguments are heuristic — mainly relying on decoupling the generator (fixed diffusion model) from the adaptive critic, rather than a principled proof.
2. The diffusion model is pretrained independently of the MARL task using random environment samples.  This means the model can only generate designs lying within the support of the pretraining distribution. If the task’s optimal environments fall outside that distribution, DiCoDe cannot discover them.
3. The benchmarks mostly test spatial layout optimisation; dynamic environment parameters (e.g., stochastic rules, non-stationary dynamics) are not explored. Competing baselines are limited — the comparison excludes recent regret-based UED and diffusion co-design methods (like ADD [1]), making it hard to quantify progress beyond the immediate predecessors.
4. DiCoDe requires three separately trained components (diffusion model, MARL policy, and environment critic) with multi-stage coordination, which significantly increases training complexity and compute demand. The paper acknowledges this but doesn’t provide quantitative training-time comparisons or scaling analysis with respect to environment dimensionality.
5. Realistic design tasks often involve multiple competing objectives (e.g., safety, energy efficiency, human preference).  The paper briefly mentions this as future work but does not experiment with or theoretically extend to multi-objective optimisation.

[1] Chung, H., Lee, J., Kim, M., Kim, D., & Oh, S. (2024). _Adversarial Environment Design via Regret-Guided Diffusion Models_. In _Advances in Neural Information Processing Systems (NeurIPS 37)_.

**Questions:**

Please answer the following questions:
1. In Eq. (6), how is $\omega$ chosen, and does the resulting trade-off preserve the optimal joint solution or bias the diffusion toward high-entropy but sub-optimal regions?
2. The paper substitutes the estimated gradients $\nabla\_{\theta} u + \omega \nabla\_{\theta} V'$ into the reverse SDE. Are these gradients unbiased estimates of $\nabla\_{\theta} \log \Lambda^{*}\_{\phi}$, and what happens when $\nabla_{\theta} V'$ is inaccurate?
3. The paper claims 66 % fewer samples. Is this reduction measured in total environment evaluations, MARL steps, or wall-clock time?
4. Does the pretrained diffusion model require retraining for each task (D-RWARE, WFCRL, VMAS), or can it generalise across design domains?
5. How sensitive is Projected Universal Guidance (PUG) to the projection operator $P_\Theta$ — does its differentiability or non-convexity affect convergence? The reverse diffusion update is derived under the assumption that gradients $\nabla_{\theta_t}$ are well-defined and continuous.  If $P_\Theta$ is non-differentiable (e.g., clipping, rounding, or nearest-valid-cell projection), then the effective sampling dynamics break the continuity assumption of the SDE approximation. If $P_\Theta$​ is non-convex, the projection may not be unique — multiple feasible projections could exist.  That can make the diffusion process stochastic in unintended ways, and might bias the trajectory of the reverse process away from the true guided distribution. In diffusion training, the convergence of score matching relies on smoothness of the forward–reverse transitions. A hard projection step effectively introduces discontinuities in the score field, which can cause unstable or biased sampling, especially if used iteratively across hundreds of timesteps. Could PUG create bias if projection removes valid but rare designs?

Potential typo:

Is there any typo in Eq. (8)? I cannot find any term inside $\nabla_{x}(\cdot)$ is with respect to the variable $x$.

---

> ### Author Response · Authors · 2025-11-21
>
> Thank you for your comments, and we appreciate the thought and insight that has gone into them. We are encouraged that you found our framework bridges the co-design closer to real-world deployment, with credible and generalisable empirical results that demonstrates the necessity of our contributions.
>
> **W1: Theoretical analysis of convergence or stability in the co-design process**
>
> We agree that formal theoretical analysis is important. However, general MARL suffers from non-stationarity as agents adapt to each other, a challenge exacerbated by a non-stationary environment. Formal convergence guarantees to global optima remain an open question not just for DiCoDe, but for MARL algorithms and bi-level optimization in general. The challenges of convergence in bi-level optimisation are discussed further in [Amir et al.], Appendix P2.
>
> * Michael Amir, Matteo Bettini, & Amanda Prorok. (2025). When Is Diversity Rewarded in Cooperative Multi-Agent Learning? https://www.arxiv.org/abs/2506.09434.
>
> **W2: The pretrained diffusion model can only generate designs lying within the support of the pretraining distribution. If the task's optimal environments fall outside the distribution, DiCoDecannot discover them.**
>
> In our experiments, we assume a uniform distribution over all valid environments as the pretraining distribution. The optimal environment must therefore be in the pretraining distribution by definition, thus we believe this does not represent a limitation of our method.
>
> Optionally, we could pretrain on a non-uniform distribution. For example, using heuristics derived by domain experts. We believe this represents a potential advantage of our framework in exploiting existing knowledge, and is a promising path for future work to explore.
>
> **W3: The benchmarks mostly test spatial layout optimisation; dynamic environment parameters are not explored. Competing baselines are limited, excluding recent regret-based UED methods like ADD**
>
> We respectfully clarify that we do compare with ADD [Chung et al.]. It is included in Appendix A.3, and critically, the DiCoDe-ADD ablation in Table 1 directly assesses the contribution of PUG against the guidance method in ADD. Our results show PUG is crucial to the empirical success of DiCoDe in the co-design domain (e.g., 57% improvement over DiCoDe-ADD in D-RWARE).
>
> We cannot directly compare DiCoDe with UED methods (e.g., regret-based UED) because Unsupervised Environment Design is a fundamentally different paradigm with different objectives (adversarial curriculum generation vs. cooperative performance maximization).
>
> Our primary objective is to establish diffusion models as a frontier for environment generation in agent-environment co-design. Prior works in this specific field are also typically concerned with static environments. We agree that dynamic environment parameters are an exciting topic for future work but fall outside the scope of this foundational paper.
>
> * Chung, H., Lee, J., Kim, M., Kim, D., & Oh, S. (2024). Adversarial Environment Design via Regret-Guided Diffusion Models. In The Thirty-eighth Annual Conference on Neural Information Processing Systems.
>
> **W4: DiCoDe significantly increases training complexity and compute demand**
>
> We argue that the most critical metric for real-world application is sample efficiency, as sample collection is often significantly costlier than computation. DiCoDe achieves sample-efficiency far beyond that of the prior state-of-the-art (RL), achieving higher performance with a fraction of the samples taken. In practical real-world tasks, sample collection can be a significantly higher fraction of the wall-clock time than  simulated benchmarks.
>
> Training time (ignoring diffusion model pretraining, which was not a substantial time-sink in our workflow),  DiCoDe took roughly 90% more wall-clock time than no co-design, but 30% less time than RL, normalised to the number of MARL steps. We do not report this time because a) the measured time is conditioned on the different servers used to run experiments, and b) the implementation of both DiCoDe and RL is not optimised. Hence, these numbers do not reflect a production-ready system.
>
> We provide scaling analysis with respect to environment dimensionality in Figure 4.

---

> ### Author Response · Authors · 2025-11-21
>
> **Q1: How is the weighting factor for environment guidance chosen?**
>
> $\omega$ is a hyperparameter that adds flexibility to the training process. In our experiments, we tune $\omega$ by evaluating the score of a generated environment using a fixed environment critic and applying empirical judgement. We agree that the selected values of $\omega$ may not be the most optimal. Akin to the exploration coefficient in RL, given sufficient compute, our method would benefit from a full sweep over values of $\omega$.
>
> We find that $\omega$ can optionally be annealed from a low initial value (for exploration over a wide range of environments and escaping local optima) to a high value at the end towards optimal environments, and use linear annealing in the windfarm experiments.
>
> **Q2: Estimated gradients of the score function**
>
>
> Thank you for the excellent question. The amount of bias present in the environment critic training target depends primarily on the bias present in the agent critic. In practice, we found overfitting the environment critic early in the training run a larger concern, where the inaccurate guidance signals resulted in sub-optimal environment-policy pairs. As mentioned previously, annealing $\omega$ mitigates this issue.
>
> **Q3: The paper claims 66 % fewer samples. Is this reduction measured in total environment evaluations, MARL steps, or wall-clock time?**
>
> This is measured as a reduction in the number of environment evaluations, keeping the number of MARL steps (policy updates) equal between DiCoDe and all comparisons. Compared to RL, which runs additional sample collections in each MARL step to collect training data solely used in training the environment generator, this demonstrates the sample-efficiency advantage of our method.
>
> **Q4: Does the pretrained diffusion model require retraining for each task (D-RWARE, WFCRL, VMAS)?**
>
>  In our implementation, we pre-train a diffusion model for each task. Training generalist diffusion models is in the field of geometric deep learning (e.g. using a GNN to train diffusion models on different sized maps), and out of the scope of this paper. We found that the time (D-RWARE) taken to pretrain the diffusion model was not substantial, taking less than 10 hours on single GPU consumer grade hardware whereas training runs could take up to 48 hours.
>
> **Q5: PUG sensitivity to non-differentiable and non-convex spaces**
>
> Thank you for the excellent question, you are correct from a theoretical standpoint we do not provide guarantees of the characteristics of PUG. We would like to point you to the predecessor work proposing PDM [Christopher et al.] which has a sound theoretical basis and shares common components with PUG.
>
> **Non-differentiability**: In PDM, the optimization is well-formed and guaranteed to converge for non-differentiable projections under convex constraints. Crucially, this proof relies on multiple iterative projection steps at each diffusion timestep, which resembles recurrent steps in PUG. We speculate recurrent steps are a factor in the empirical difference between applying PDM on DDIM naively, and PUG. Used iteratively across hundreds of diffusion timesteps, projections keep the sample within the support of the training dataset where the model provides more accurate score estimates, rather than going out-of-distribution. Delving deeper into diffusion inference is out of the scope of this paper, but could be interesting follow-up work.
>
> **Non-convexity**: Although neither PDM or PUG guarantee convergence under non-convex conditions, empirical results demonstrate these are highly practical strategies. In particular, note the success of PUG in the image space for D-RWARE, where the space of valid environments is highly disconnected and non-convex.
>
> * Christopher, J., Baek, S., & Fioretto, N. (2024). Constrained synthesis with projected diffusion models. Advances in Neural Information Processing Systems, 37, 89307–89333.
>
> **Q6: Typo in Eq.8**
>
> Thank you for catching this, we have applied the relevant correction.

---

> > ### Comment · Reviewer_SXPT · 2025-11-22
> >
> > I appreciate your concrete response. I am satisfied with most of answers. Although some of them have not been directly resolved, I can understand that it cannot be immediately addressed for the time being.

---

### Official Review · Reviewer_JUSx · 2025-10-30

**Soundness:** 3
**Presentation:** 3
**Contribution:** 3
**Rating:** 8
**Confidence:** 4

**Summary:**

This paper develops an algorithm combining diffusion models and multi-agent reinforcement learning (MARL) for finding an optimal design and agent strategy to maximize the utility. The algorithm has two main parts: the first part is the so-called projected universal guidance to generate a valid environment via projection, then they learn an environment critic by incorporating an estimate of the agent critic function for sample efficiency. Empirical results across warehouse logistics, wind farm control, and multi-agent navigation show substantial performance and scalability improvements.

Overall, I believe this paper has a good theoretical background and has good methodological improvements in terms of practical performance, and I think the equations and algorithms are clear, and the numerical results look convincing. Therefore, I recommend accepting the paper.

**Strengths:**

I think introducing diffusion models for optimizing the environment and policy for a multi-agent reinforcement learning problem is a good strength. The authors also introduce 2 key insights, namely the projection operator in generating the environment, and also using the agent critic estimates while optimizing for the environment critic function.

**Weaknesses:**

I do not see a major weakness for this paper.

**Questions:**

Could you explain the projection operator's operation more clearly? It may be possible that for certain applications, it is challenging to compute a feasible configuration or identify the "closest" configuration based on the problem setup.

You mentioned PDM, and stated that PUG is more efficient because it does not require the solution to be feasible at all times. How does the projection operator specifically help here?

I understand the motivation behind introducing the agent critic estimate into the cost function for optimizing the environment critic. How does that compare to the shared experience actor-critic approaches for MARL?

I would also appreciate more insight into how the set \Theta is chosen for the samples. The authors mention that it's a FIFO queue, but is it possible to have a biased estimate if the agent critic is not sufficiently converged with respect to the environment?

---

> ### Author Response · Authors · 2025-11-21
>
> Thank you for your insightful comments. We are encouraged that you found our methodological improvements introducing diffusion models for optimising the environment convincing, and our equations and algorithms clear.
>
> **Q1. Clarifications on the projection operator**
>
> Yes, the projection operator is a domain specific construct, and can be potentially challenging to compute for unique constraints. In practice, projection onto the feasible set (often referred to as post-processing projection) is well-established for a wide range of problems, and is far more flexible than those examined in prior agent-environment co-design tasks. For example, ADD [Chung et al.] assume the entire domain is valid.
>
> The key insight to PUG from PDM is the usage of the expected clean image (Eq. 9). We apply $\mathcal{P}$ to $x_0$. This does not enforce $x_t$ to be constrained after adding noise back into $x_0$ in the process of universal guidance.  In contrast, PDM projects the constraint onto $x_t$ directly.
>
> We remark the primary motivation for incorporating the expected clean image is not due to the relaxed constraints compared to PDM, but rather, it is impractical to train a noise-conditioned environment critic.
>
> * Chung, H., Lee, J., Kim, M., Kim, D., & Oh, S. (2024). Adversarial Environment Design via Regret-Guided Diffusion Models. In The Thirty-eighth Annual Conference on Neural Information Processing Systems.
>
> **Q2: Comparing critic distillation with shared experience actor-critic approaches**
>
> This is an insightful comparison. Our approach (critic distillation) and Shared Experience Actor-Critic (SEAC) [Christianos et al.] both aim to improve sample efficiency through knowledge sharing, but they target fundamentally different aspects of the learning problem.
> SEAC is a method of *agent-to-agent* knowledge transfer. It shares experiences *between agents* to improve MARL policy optimisation. It allows agents to learn from the trajectories of others, using off-policy corrections (importance sampling) to combine gradients. This accelerates exploration and helps synchronise agent learning rates within the MARL loop.
> In contrast, DiCoDe's critic distillation is a method of *agent critic-to-environment critic* knowledge transfer. We share knowledge from the agents (via the agent critic) to the *environment optimizer* (via the environment critic). The goal is not to improve the agent policies directly, but to provide a stable, dense, and up-to-date target for the environment generator. This specifically mitigates the policy shift inherent in the bi-level co-design optimization.
> Crucially, these mechanisms are orthogonal. DiCoDe is a framework for co-design (optimizing environment and agents), whereas SEAC is a MARL algorithm focused solely on agent training. SEAC could readily be substituted as the underlying MARL backbone within DiCoDe.
>
> * Christianos et al., Shared Experience Actor-Critic for Multi-Agent Reinforcement Learning. https://arxiv.org/pdf/2006.07169
>
> **Q3: Insight into environment critic training dataset**
>
> Each time an environment is sampled in a trajectory, it is added to the queue. The max size of the queue is a hyper-parameter chosen by the practitioner. We note when tuning experiments, performance is robust to this hyperparameter with critic distillation, whereas the ablation DiCoDe-MC suffers when the size of the queue is small.
>
> We don’t present theoretical guarantees of convergence, as this is an open problem for not just co-design, but MARL with function approximation in general. We infer your question refers to two sources of estimation error, and we explain how they may be mitigated in practice.
> - Moving target from the changing environment generator. This bias could be removed by avoiding the queue and sampling θ directly from the up-to-date generator. However, we empirically found negligible differences in training when using the FIFO queue.
> - Prediction error in the agent critic. Indeed, we found that DiCoDe could overfit to suboptimal environments if the environment critic fits to noisy agent critic predictions early in training (insufficient exploration). The hyperparameter $\\omega$ (Eq. 6) allows us to mitigate this using an annealing schedule: we explore a diverse range of environments early on (avoiding overfitting of either critic), and then exploit the best environments once the critics are reasonably trained.

---

### Author Response · Authors · 2025-11-21

We thank the reviewers for their thoughtful engagement with our work and their constructive feedback. We are encouraged that reviewers found our framework "bridges the co-design closer to real-world deployment" (SXPT) with "convincing" methodological improvements (JUSx) and "credible and generalisable empirical results" (SXPT). We have made the following changes in the revised manuscript:

- **Included additional references**, including recent related work on the co-design. We have added Section A.4 in the Appendix to provide a clearer background on the use of diffusion models in reinforcement learning, including those beyond environment generation.
- **Added additional environment visualisations** (Figure 4, 8, 9) to aid in qualitative analysis of our method, and a short discussion in Section 5.
- **Reorganised Appendix**. Improved formatting on the additional diffusion model background
- **Amended typos**

We hope our comments address the reviewer's concerns, and welcome any further questions.

---

### Comment · Area_Chair_b2Hi · 2025-11-25
**Please read the rebuttal and respond**

Dear reviewers,

Now that the author responses are in, could you please take a look at them and see if they address your concerns adequately?

Thank you very much.

Best,
AC

---

### Meta-Review · Area_Chair_7N3v · 2026-01-04

**Summary:**

The reviewers acknowledged the effectiveness of integrating diffusion models into the co-design loop and the empirical improvements over specific baselines in certain benchmark tasks. However, concerns remain regarding the overall complexity of the proposed pipeline, as well as the generality and widespread adoption of the selected benchmark tasks.

After carefully weighing the reviews and the authors' rebuttal, the AC recommends rejection. This was a difficult decision made in the context of a highly competitive batch of submissions. While the proposed method is technically sound within its specific domain, the work is currently constrained by a scope that may restrict its immediate significance given the current form (the baselines employed are not fully representative of the broader state-of-the-art in multi-agent learning; the evaluation would benefit from inclusion of more standard benchmarks or application to more complex, widely-studied domains to better demonstrate practical utility).

**Reviewer Concerns:**

Concerns addressed: The authors provided a comprehensive rebuttal that resolved the majority of technical clarifications and misunderstandings raised by the reviewers.
- The authors provided clear explanations regarding the projection operator (PUG vs. PDM), and also distinguished their method from SEAC.
- The concern regarding missing regret-based baselines (specifically ADD) was resolved, as the authors pointed out its inclusion in the Appendix and demonstrated better performance. The authors also clarified that the pretraining distribution covers all valid environments.
- The authors clarified that the ablation study regarding critic distillation requested by the reviewer was confirmed to be already present in the paper.

Outstanding concerns: Despite the effective rebuttal, the following concerns regarding the fundamental nature of the proposed approach remain and weigh against acceptance:

- Computational complexity: The authors acknowledged a significant increase in wall-clock time compared to baselines. While the authors argue that sample efficiency is paramount, the heavy computational burden of training three coupled components restricts the method's practicality in resource-constrained scenarios. The trade-off between sample efficiency and wall-clock time is highly dependent on the specific application domain, and this limitation suggests the pipeline may be too heavy for general adoption.

**Reviewer Scores:**

- Reviewer JUSx: maintain positive
- Reviewer SXPT: maintain positive
- Reviewer rKbS: maintain negative

---

### Decision · Program_Chairs · 2026-01-26

Reject